

# Implementation and validation of a new operational wave forecasting system of the Mediterranean Monitoring and Forecasting Centre in the framework of the Copernicus Marine Environment Monitoring Service

Michalis Ravdas[1,2], Anna Zacharioudaki[1,2] and Gerasimos Korres[1],

[1] Hellenic Centre for Marine Research, P.O. Box 712, 19013 Anavissos, Hellas
[2] These authors contributed equally to this work

*Correspondence to*: Anna Zacharioudaki (azacharioudaki@hcmr.gr)

**Abstract.**

Within the framework of the Copernicus Marine Environment Monitoring Service (CMEMS) an operational wave forecasting system for the Mediterranean Sea has been implemented by the Hellenic Centre for Marine Research (HCMR) and evaluated through a series of pre-operational tests and subsequently for one full year of simulations (2014). The system is based on the WAM model and it has been developed as a nested sequence of two computational grids to ensure that occasional remote swell propagating from the North Atlantic is correctly entering into the Mediterranean Sea through the

Gibraltar Strait. The Mediterranean model has a grid spacing of $1/24^{\circ}$. It is driven with 6-hourly analysis and 5-days forecast 10m ECMWF winds. It accounts for shoaling and refraction due to bathymetry and surface currents which are provided in off-line mode by CMEMS. Extensive statistics on the system performance have been calculated by comparing model results with in-situ and satellite observations. Overall, the significant wave height is accurately simulated by the model while less accurate but reasonably good results are obtained for the mean wave period. In both cases, the model performs optimally at

offshore wave buoy locations and well-exposed Mediterranean sub-regions. Within enclosed basins and near the coast, unresolved topography by the wind and wave models and fetch limitations cause the wave model performance to deteriorate. Model performance is better in winter when the wave conditions are well-defined. On the whole, the new forecast system provides reliable forecasts. Future improvements include data assimilation and higher resolution wind forcing.





# 1   Introduction

In recent years the requirements of marine industry for real-time wave forecasts have increased substantially. Various sectors such as for example the maritime transport, shipping and offshore mineral industry require accurate wave forecasts in order to secure operations at sea, save fleet fuel consumption using more accurate routing and prevent from potential ship and platform oil spill drift, whilst a detailed wave information along coastal regions is crucial for coast guards and port authorities as there is the need to anticipate wave conditions that could interfere in ships arriving and leaving the harbours. Furthermore, waves forecasting provides significant advantages for the offshore wind and the wave energy so as to schedule installation and maintenance activities, to define control strategies according to the predominant wave conditions, or to plan for storm events. Scientifically, waves are a bridge between the ocean and the atmosphere, playing a key-role in the air-sea interaction and are also an important mixing agent with an active role in erosion and re-suspension processes.

The dramatic increase in computing power and the enhanced understanding of the physical processes responsible for wave generation, evolution and dissipation have resulted in third-generation wave models which use first principles in the integration of an action or energy balance equation (Tolman, 1992) based on the sophisticated physics pertaining to wave generation, propagation and decay mechanisms.  Wave models like WAM (WAMDI Group, 1988; Komen et al., 1994), WAVEWATCH III (Tolman, 1991, 1999) and SWAN (Booij et al., 1999) are being used by many meteorological and oceanographic operational centres and have been reasonably successful in operational wave predictions at the global, regional and coastal scales. One of the pioneers in the implementation and the development of wave analysis and forecast systems is the European Centre for Medium Weather Forecast (ECMWF) which provides since 1992 daily mid-range global waves forecasts up to ten days ahead. With time, more centres (e.g. UK Met Office, the Service Hydrographique et Océanographique de la Marine (SHOM) and several others) began to use state-of-the-art third generation  numerical wave models in operational forecasting. In addition an international wave forecasts inter-comparison project was established (Bidlot et al., 2007), coordinated by the ECMWF, to evaluate forecasts quality and performance and identify areas of potential improvement (Breivik et al., 2015).

The EU-funded Copernicus Marine Environmental Monitoring Service (CMEMS) driven by the requirements of a large user panel in need of wave information in all ocean basins enriched its portfolio since early 2017 with the provision of open, cost free and quality controlled wave products for the global ocean and the European regional seas. CMEMS is based on a strong European partnership with more than 50 marine operational and research centres in Europe involved in the marine monitoring and forecasting services and their evolution providing a wide range of marine products of social and environmental value such as ocean currents, temperature, salinity, sea level, pelagic biogeochemistry and waves. The backbone of the CMEMS relies on a Central Information System (CIS) and an architecture of production centres inherited from the MyOcean projects both for observations (Thematic Assembly Centres - TACs) and modelling/assimilation (Monitoring and Forecasting Centres - MFCs). The MFCs are distributed according to the marine area covered and generate model-based products including analysis of the current situation, forecasts of the situation a few days in advance and the





provision of retrospective data records (re-analysis). Detailed information on the systems and products are on CMEMS web site: http://marine.copernicus.eu/. The Mediterranean Sea (Med) MFC is composed of INGV (Italy), HCMR (Greece), OGS (Italy) and the Euro-Mediterranean Centre on Climate Change (CMCC, Italy). It is an expert consortium with profound expertise in the Mediterranean phenomenology and dynamics, from waves to currents and biogeochemistry and provides regular and systematic information about the physical state of the ocean and the dynamics of the marine ecosystem of the basin.

In this study we present the wave component of Med-MFC, a high resolution operational wave forecasting system (hereafter called Med-waves), which has been developed by HCMR and provides to the general public through the CMEMS portal daily accurate products – wave simulations and 5-day forecasts- of the wave environment of the Mediterranean Sea. The system employs the WAM Cycle 4.5.4 (Gunther and Behrens, 2012), a modernized and improved version of WAM model. In this study, a modelling system consisting of a nesting sequence of two computational grids (North Atlantic and Mediterranean) has been developed with the fine grid model covering the Mediterranean Sea and the coarse grid model, the North Atlantic. A nesting approach of this kind enables to properly simulate the effect of the remotely generated Atlantic swell into the Mediterranean Sea as it passes through the Gibraltar Strait. In fact, Cavaleri and Sclavo (2006) pointed out that the narrow strait of Gibraltar affects appreciably the wave climate in the close-by area of the Alboran Sea and it is often neglected in wave modelling systems of the Mediterranean Sea. Moreover, the system incorporates off-line coupling with general circulation models (CMEMS Global and Mediterranean analysis and forecast systems) to provide surface currents for wave refraction to both nests of the modelling system. Refraction due to surface currents impact to some extend the wave spectrum. Its impact on improving the wave forecast is addressed by Osuna and Wolf (2005) and Clementi et al. (2013). Thus, while in the last years many major operational centres have already developed and implemented wave forecast systems for the entire Mediterranean, none of these takes into account the sensitivity of Mediterranean wave dynamics to the nesting with the Atlantic incorporating at the same time the surface currents effect to wave refraction. In addition the system offers an extended set of freely available wave products and has a spatial and spectral resolution high enough to describe with sufficient accuracy the wind wave dynamics over the Mediterranean basin.

Despite the diversity of wave generation models as well as of atmospheric models, the good quality of the short-term wave forecasts for the Mediterranean Sea is still a difficult task due to the large spatial and temporal variability of the surface wind field over the basin. Wind-wave models are very sensitive to wind field variations which result in one of the main source of errors in wave predictions. The sensitivity of wave model prediction to variations in wind forcing fields has been studied by several authors (Komen et al., 1994; Teixeira et al., 1995; Holthuijsenet et al., 1996; Ponce and Ocampo-Torres, 1998). This is particularly true for the Mediterranean where the limited contribution of swell to the wave spectrum makes the regional wind conditions the most important factor in determining the local wave state. Bearing in mind this context, the complex structure of the Mediterranean Sea due to the presence of large mountainous islands, protruding peninsulas, jagged coastlines and sharp orography gradients, deeply influence the wind and wave dynamics, especially close to the coast, making the local forecast particularly challenging. Many authors have highlighted the fact that forecasted winds





in the Mediterranean are not as accurate as in the open ocean (Cavaleri and Bertotti, 2003; 2004; Signell et al., 2005; Ardhuin et al., 2007; Bolanos-Sanchez et al., 2007) and advocate the necessity of further improvement on the wind field quality as well as the increase of its spatial resolution especially in enclosed basins such as the Mediterranean Sea (Cavaleri and Bertotti, 2004; 2009; Bentamy et al., 2007; Lionello et al., 2008). Hence, the selection of an appropriate wind data set is

a vital step in the wave modelling of the Mediterranean Sea. As such, the Med-waves system in particular uses the analysis and forecast wind fields produced by ECMWF Integrated Forecasting System (IFS). The quality and appropriateness of ECMFW 10m winds for wave simulations and forecasting in different areas has been demonstrated by many studies. On a global scale, repetitive statistics have shown that the ECMWF products are, and have been for a long while, the best ones in the world (Bertotti et al., 2011). However, for semi-enclosed basin, the quality of ECMWF wind fields decrease and the

wave model underestimates the high wave heights because of the underestimation of high wind speeds (Cavaleri and Bertotti, 2004; Signell et al., 2005; Cavaleri and Scalvo, 2006; Saket et al., 2013) and/or overestimates the lower ones because of the overestimation of low wind speeds by ECMWF (Moeini et al., 2010). In our system, this underestimation of wind is compensated by reducing the energy loss due to whitecapping, performing a fine tuning of the free parameters of the dissipation function.

15        We present here the first comprehensive documentation of the system and the evaluation of its accuracy over a period of one year (2014). The rest of the paper is organized as follows. A detailed description of the Med-waves modelling system is given in Section 2. Section 3 outlines the methodology followed in the model validation, while Section 4 is devoted to the validation results including both hindcast and forecast skill evaluation against in-situ and satellite observations. Finally, a summary and some concluding remarks are given in section 5.

## 2     The wave forecasting system

       As previously said, Med-waves is based on the WAM Cycle 4.5.4 wave model, a state-of-the-art third-generation wave model which is a modernized and improved version of the well-known and extensively used WAM Cycle 4 wave model (WAMDI Group, 1988; Komen et al., 1994). Cycle 4.5.4 has been released during MyWave ("A pan - European concerted and integrated approach to operational wave modelling and forecasting – a complement to GMES MyOcean

services") EU FP7 Research Project and is freely available to the entire research and forecasting community. WAM solves the wave transport equation explicitly without any presumption on the shape of the wave spectrum. Its source/sink terms include the wind input, whitecapping dissipation, nonlinear transfer and bottom friction. The wind input and whitecapping dissipation source terms of the present cycle of the wave model are a further development based on Janssen´s quasi-linear theory of wind-wave generation (Janssen, 1989; 1991). The nonlinear transfer term is a parameterization of the exact

nonlinear interactions as proposed by Hasselmann and Hasselmann (1985) and Hasselmann et al. (1985). Lastly, the bottom friction term is based on the empirical JONSWAP model of Hasselmann et al. (1973).





The Med-waves set-up includes a coarse grid domain with a resolution of 1/6º covering the North Atlantic Ocean (NA) from 75ºW to 10ºE and from 10ºN to 70ºN and a nested fine grid domain with a resolution of 1/24º covering the Mediterranean Sea from 18.125ºW to 36.2917ºE and from 30.1875ºN to 45.9792ºN. The areas covered by the two grids are shown in Fig. 1 which is a schematic of the Med-waves system. The bathymetric map has been constructed using the

GEBCO 30 second bathymetric data set (GEBCO, 2016) for the Mediterranean Sea model and the ETOPO2 2-minute bathymetric data set (NGDC, 2006) for the North Atlantic model.

The Mediterranean Sea model receives from the North Atlantic model full wave spectrum at hourly intervals at its Atlantic Ocean open boundary. The latter model is considered to have all of its four boundaries closed with no wave energy propagation from the adjacent seas. Because of the wide geographical coverage of the North Atlantic model, the

consideration of closed boundaries does not affect the swell propagation towards the open boundary of the Mediterranean model which is the main interest of this nesting approach.

The wave spectrum is discretized using 32 frequencies, which cover a logarithmically scaled frequency band from 0.04177 Hz to 0.8018 Hz at intervals of $\frac{df}{f} = 0.1$, and 24 equally spaced directions (15° bin size).

The Mediterranean model runs in shallow water mode considering wave refraction due to depth and currents in

addition to depth induced wave breaking. The North Atlantic model runs in deep water mode with wave refraction due to currents only. The North Atlantic model additionally considers wave energy damping due to the presence of sea ice.

Following ECMWF (ECMWF, 2015), the tunable whitecapping dissipation coefficients $C_{dis}$ and $\delta$ have been altered from their default values. Specifically, the values of $C_{dis} = 1.33$ ($C_{dis} = 2.1$ default) and $\delta = 0.5$ ($\delta = 0.6$ default) have been adopted. The aim of this tuning is to produce results which are in good agreement with data on fetch-limited growth

and with data on the dependence of the surface stress on wave age.

The atmospheric forcing of Med-waves consists of 10 m above sea surface wind fields at 1/8° horizontal resolution originating from ECMWF IFS (ECMWF, 2016). The ECMWF IFS is an operational global meteorological forecasting model that is presently one of the predominant synoptic-scale medium-range models in general use worldwide. It is hydrostatic, two-time-level, semi-implicit and semi-Lagrangian and applies spectral transforms between grid-point space and

spectral space. The quality of the ECMWF forecasts is regularly evaluated (Haiden et al., 2016). Sea ice coverage fields are also obtained from ECMWF IFS at the same horizontal resolution as the wind fields.

Surface currents forcing is accounted for in Med-waves. The Mediterranean Sea model is forced by surface currents obtained from the physical forecasting system of the CMEMS Med-MFC at 1/16° horizontal resolution (CMEMS, 2016a) and the North Atlantic model by surface currents obtained from the physical forecasting system of the CMEMS Global-MFC

at 1/12° horizontal resolution (CMEMS, 2016b). Both physical forecasting systems are based on the Nucleus for European Modelling of the Ocean (NEMO) ocean physics model which is a state-of-the-art free-surface primitive equation model (Madec, 2008). NEMO is free software used by a large community.





Med-waves is run once per day starting at 12:00:00 UTC. It produces 5-day forecast fields initialized by a 1-day hindcast. The wave hindcast is forced by 6-hourly analysis wind fields and daily averaged analysis current fields. The 5-day forecast is forced by 3-hourly forecast wind fields for the first 3 days and 6-hourly forecast wind fields for the rest of the forecast cycle. Daily averaged forecast currents are used over the entire wave forecast. Sea ice coverage fields are updated at daily frequency and remain constant during the forecast cycle.

Med-waves generates hourly wave fields over the Mediterranean Sea at 1/24° horizontal resolution. These wave fields correspond either to wave parameters computed by integration of the total wave spectrum or to wave parameters computed using wave spectrum partitioning. In the latter case the complex wave spectrum is partitioned into wind sea, primary and secondary swell. Wind sea is defined as those wave components that are subject to wind forcing while the remaining part of the spectrum is termed swell. Wave components are considered to be subject to wind forcing when:

$$c \leq 1.2 \times 28 u_* \cos(\vartheta - \varphi) \qquad (1)$$

where $c$ is the phase speed of the wave component, $u_*$ is the wind friction velocity, $\theta$ is the direction of wave propagation and $\varphi$ is the wind direction. As the swell part of the wave spectrum can be made up of different swell systems with quite distinct characteristics it is further partitioned into the two most energetic wave systems, the so called primary and secondary swell. Swell partitioning is done following the method proposed by Gerling (1992) which finds the lowest energy threshold value at which upper parts of the spectrum get disconnected with the process repeated until primary and secondary swell is detected.

Total spectrum and partitioned wave parameters produced by Med-waves and disseminated though CMEMS include: spectral significant wave height (Hm0), spectral moments (-1,0) wave period (Tm-10), spectral moments (0,2) wave period (Tm02), wave period at spectral peak / peak period (Tp), mean wave direction from (Mdir), wave principal direction at spectral peak, stokes drift U, stokes drift V, spectral significant wind wave height, spectral moments (0,1) wind wave period, mean wind wave direction from, spectral significant primary swell wave height, spectral moments (0,1) primary swell wave period, mean primary swell wave direction from, spectral significant secondary swell wave height, spectral moments (0,1) secondary swell wave period, mean secondary swell wave direction from.

## 3 Validation framework

Med-waves has been validated against in-situ and satellite observations, focusing on its performance in the Mediterranean Sea. Model output and observations corresponding to the year 2014 have been compared, focusing on the fundamental wave parameters of significant wave height, Hs, and mean wave period, Tm.

In-situ measurements of Hs and Tm for 2014 were extracted from the Copernicus In Situ Thematic Assemble Centre (INS-TAC), a component of CMEMS which aims at providing a research and operational framework to develop and deliver in situ observations and derived products based on such observations. Hs measurements from 32 wave buoys within the Mediterranean Sea were available in the examined year. Figure 2 depicts their location and unique ID code. Tm



measurements were available from a sub-set of the depicted buoys which excludes all the buoys offshore from the Italian coastline. To collocate model output and buoy measurements, in space, model output was taken at the grid point nearest to the buoy location. In time, buoy measurements within a time window of ± 1 h from model output times at 3-h intervals (0, 3, 6 ..., etc) were averaged. Prior to model-buoy collocation, the in-situ observations were filtered so as to remove those values

accompanied by a bad quality flag (Quality Flags included in the data files provided by the INS-TAC). After collocation, visual inspection of the data was carried out, which led to some further filtering of spurious data points. In addition, Tm data below 2 sec were omitted from the statistical analysis, since 0.5 Hz (T = 2 sec) is a typical cut-off frequency for wave buoys.

Satellite observations of significant wave height, Hs, and wind speed, U10 (used to obtain Hs-U10 quality associations), for year 2014 were obtained from a merged altimeter wave height database setup at CERSAT - IFREMER

(France). This database contains altimeter measurements that have been filtered and corrected (Queffeulou and Croizé-Fillon, 2013). Here, measurements from 3 satellite missions, the Jason-2, Cryosat-2, and Saral, were used. To collocate model output and satellite observations the former were interpolated in time and space to the individual satellite tracks. For each track, corresponding to one satellite pass, along-track pairs of satellite measurements and interpolated model output were averaged over ~50 km (0.5°) grid cells, centered at grid points of the forcing wind model (0.125° x 0.125°). This

averaging is intended to break any spatial correlation present in successive 1 Hz (~7 km) observations and/or in neighboring model grid output (Queffeulou, personal communication).

Metrics that are commonly applied to assess numerical model skill and are in alignment with the recommendations of the EU FP7 project MyWave (MyWave, 2014) have been used to qualify the Med-waves system within the Mediterranean Sea. These include the RMSE, BIAS, Scatter Index (SI), Pearson Correlation Coefficient (CORR), and best-fit Slope

(SLOPE). The SI, defined here as the standard deviation of errors (model - observations) relative to the observed mean, being dimensionless, is more appropriate to evaluate the relative closeness of the model output to the observations at different locations compared with the RMSE which is representative of the size of a 'typical' error. The SLOPE corresponds to a best-fit line forced through the origins (zero intercept). In addition to the aforementioned core metrics, merged Density Scatter and Quantile-Quantile (QQ) plots are provided. Metrics are computed for the Mediterranean Sea as a whole, for the

individual wave buoy locations shown in Fig. 2 and for 17 sub-regions from which 1 is in the Atlantic Ocean and 16 in the Mediterranean Sea (Fig. 3): (atl) Atlantic, (alb) Alboran Sea, (swm1) West South-West Med, (swm2) East South-West Med, (nwm) North West Med, (tyr1) North Tyrrhenian Sea, (tyr2) South Tyrrhenian Sea, (adr1) North Adriatic Sea, (adr2) South Adriatic Sea, (ion1) South-West Ionian Sea, (ion2) South-East Ionian Sea, (ion3) North Ionian 3, (aeg) Aegean Sea, (lev1) West Levantine, (lev2) North-Central Levantine, (lev3) South-Central Levantine, (lev4) East Levantine. All metrics are

evaluated over a period of 1 year (2014). In addition, metrics associated with the full Mediterranean Sea are evaluated seasonally.



## 4    Validation Results

### 4.1    Hindcast significant wave height

#### 4.1.1    Comparison with in-situ observations

Table 1 shows results of the comparison between hindcast Hs (model data) and in-situ observations (reference data),
for the Mediterranean Sea as a whole, for the entire year of 2014 and seasonally. In the table, "Entries" refers to the number
of model-buoy collocation pairs, i.e. to the sample size available for the computation of the relevant statistics, $\overline{R}$ is the mean
reference value, $\overline{M}$ is the mean model value, STD R and STD M are the standard deviations of the reference and model data
respectively. The remaining quantities are the qualification metrics defined in the previous section. Figure 4 is the respective
merged QQ-Scatter plot for the full 1-year period. In the figure, the QQ-plot is depicted with black crosses. Also shown are
the best fit line forced through the origin (red solid line) and the 45° reference line (red dashed line).

Table 1 shows that the typical error (RMSE) varies from 0.17 m in summer to 0.25 m in winter. However, the
scatter in summer (0.26) is about 2% higher than the scatter in winter (0.24) whilst a lower correlation coefficient is
associated with the former season. This suggests that the model follows better the observations in 'stormy' conditions, with
well-defined patterns and higher waves. A similar conclusion has been derived by other studies (Cavaleri and Sclavo, 2006;
Ardhuin et al., 2007; Bertotti et al., 2013) with respects to wind and wave modelling performance in the Mediterranean Sea.
Alike summer, autumn is characterized by lower mean wave height, higher scatter and lower correlation coefficient
compared to winter and spring. Relative BIAS (BIAS/$\overline{R}$) has an approximate variation of 2.5% (spring) to 5% (autumn) and
is always negative. Accordingly, SLOPES are also below unity with small variation between seasons. These values are
indicative of an underestimation of the wave height in the Mediterranean Sea by the model, a result which is in agreement
with the results of a number of operational or pre-operational models for the Mediterranean Sea (e.g. Bidlot, 2015; Donatini
et al., 2015) and is linked to an underestimation of the wind speed by the ECMWF forcing wind model (see Fig. 8). Overall,
the spring statistics are the ones closest to the year-long statistics for the Mediterranean Sea.

Figure 4 depicts the pattern of the agreement between hindcast and observed Hs for different Hs value ranges. The
figure reveals that the Hs underestimation by the model is mainly occurring for wave heights below 4 m and is rather small.
It also shows a QQ-plot that is really close to the reference line over most of the Hs range observed, which means that waves
of a specific wave height have a very similar probability of occurrence in the hindcast and in the observations. 'Outliers' are
present in the scatter plot, for example, a number of measured waves of 2-4.5 m height are not simulated by the model (not
enough evidence was found to remove the depicted outliers from the calculation of the statistics as faulty). These values are
likely to have introduced some bias to the computed statistics and, clearly, have led to a greater SI.

Table 2 shows results of the comparison between hindcast Hs and in-situ observations for each of the wave buoys
depicted in Fig. 2 (buoys listed from west to east). The table reveals that RMSE varies from 0.16 m to 0.44 m with the
highest values (> 0.3 m) observed in the North Adriatic Sea at buoy location 61218 (0.44) and offshore the French coastline



at buoy location 61021 (0.35). Both locations are associated with poor overall qualification metrics with location 61218 in particular displaying the poorest overall statistics. Thus, SI, which varies between 0.17 (61197) and 0.53 (61218), also obtains its highest values − indicating a poorer model performance − at locations within the North Adriatic Sea. Locations SARON and 61021 follow with values of 0.4 and 0.32 respectively. In general, SI values above the mean value for the whole

Mediterranean Sea (0.25) are obtained at all wave buoys located near the coast that are sheltered by land masses on their west (e.g. western French coastline, eastern part of Corsica and eastern part of Italy). With respects to the Italian buoys, this result is in close agreement with Cavaleri and Sclavo (2006) who found that the performance of the operational ECMWF wave model deteriorated at those Italian wave buoys facing East. This is because the resolution of the forcing wind model is not capable of well reproducing the fine interaction between the prevailing north-northwesterly winds in the northern

Mediterranean Sea with the complex orography sheltering the northern Mediterranean coastline. An underestimation of wind speeds and consequently of wave heights (also the case herein) is commonly observed at such locations (Ardhuin et al., 2007). In addition, the buoys are often located only a few kilometers from the coastline, thus, in these conditions, i.e. when the wind is blowing from the coast, the approximation of the wave model grid size can lead to non-negligible fetch differences. Similar SI values are found within enclosed basins characterized by a complex topography such as the Adriatic

and Aegean Seas. In general, the more close the location to the coastline (e.g. 61187) and/or the more complex the surrounding topography (e.g. SARON) the poorer the model performance expected. As explained in several studies (e.g. Cavaleri and Sclavo, 2006; Bertotti et al., 2013; Zacharioudaki et al., 2015), in these cases, the spatial resolution of the wave model is not adequate to resolve the fine bathymetric features whilst, as mentioned above, the spatial resolution of the forcing wind model is incapable to reproduce the fine orographic effects, introducing errors to the wave hindcast. The

correlation coefficient (CORR) largely follows the pattern of variation of the SI. It ranges from 0.79 (61218) in the Adriatic Sea to 0.98 (61213) at a location west of Sardinia which is well exposed to the prevailing north-westerly winds in the region. BIAS varies from -0.13 m at location 61218 in the Adriatic Sea to 0.11 m at location 61021 offshore from the French coastline. It is mostly negative indicating an underestimation of the observed wave height by the model, with positive BIAS observed at only 6 out of the 32 buoy locations examined. In most of the cases of relatively high positive BIAS, this is

because the wave model resolution has missed or has partly captured important bathymetric features in the surroundings of the relevant locations, thus, missing or reducing the shadowing effects produced by these features. For example, shoreward buoy 61021 where the largest positive BIAS is observed there are few small islands that are almost entirely missed by the model. Also, at buoy 61221, similarly to Bertotti et al. (2013), the coastal geometry is not well represented by WAM. Bertotti et al. (2013) state that the buoy position is exposed to the easterly waves more than is actually the case, leading to

the observed overestimate by the model. Similar conclusions stand for SARON buoy in the Aegean Sea. Overestimation at buoy 61198 in the Alboran Sea is part of a general overestimation of the wave heights in the Atlantic and Alboran regions as it will be seen later in the comparison with the satellites. In accordance with BIAS, best-fit slopes (SLOPE) vary from 0.77 at buoy location 61218 in the Adriatic Sea to 1.1 at buoy location 61021 offshore from France. SLOPE values above unity





coincide with locations of positive BIAS, otherwise they are below unity, confirming an overall underestimation of the observations by the model. In general, the pattern of variation of SLOPE is close to the pattern of variation of BIAS.

Up to now, the overall performance of the Med-waves modelling system at the different wave buoy locations has been analyzed independently of the severity of the conditions. Figure 5, similarly to Fig. 4, shows the QQ-Scattter plots of
hindcast Hs versus measured Hs at three buoy locations, exhibiting model performance over the different wave height ranges. The results at these three locations are reasonably representative of the different behaviours of the wave model at the different wave buoy locations in the Mediterranean Sea shown in Fig. 2. Thus, the top left plot shows the behaviour of the model at location 61188, offshore from the border between France and Spain, backed on the west by the Pyrenees Mountains. It is seen that model underestimation occurs throughout the measured Hs range except from the highest
percentiles of Hs where model overestimation is observed. This distribution, with a smaller or larger model underestimate and with a more or less pronounced convergence or overestimate towards the highest waves, is observed at the majority of the wave buoy locations. The bottom left plot corresponds to location 61221, south of the island of Sardinia. There, the model overestimates the observed Hs over the entire Hs range, even more so in the upper end of this range. Considerable model overestimation, mostly over the middle and higher Hs ranges, is observed at all wave buoys associated with
unresolved bathymetric features in their surroundings (e.g. 61021, SARON). Over the lower Hs range, converge or underestimate is also observed in these conditions. At SARON (not shown), the surrounding topography is highly complex, including both orographic and bathymetric effects, resulting in highly scattered data around the reference line. The right plot shows results at buoy location 61197 east of the Balearic Islands. This is a well-exposed offshore wave buoy; consequently, the behaviour of the model at this location is expected to be representative of its performance at well-exposed offshore sites.
A relatively small scatter of the data points is shown in the plot with QQ crosses and best-fit line laying close to the reference line. More specifically, the model converges to the observations for wave heights below about 2 m, somewhat underestimates the observations for wave heights between 2-4 m, and tends to overestimate Hs for higher waves. A very similar distribution is found for location 61430 west of the Balearic Islands. Other well-exposed offshore locations present QQ-Scatter patterns that are not far from the one shown for location 61197.





### 4.1.2    Comparison with satellite observations

This sub-section starts with the comparison of Med-waves hindcast Hs with satellite observations of Hs separately for each satellite. This is done for 1-year period (2014) for the full Mediterranean Sea and for the different sub-regions defined in Fig. 3. Respective results are shown in Table 3 and Fig. 6.

Table 3 shows that even though the model-satellite comparison behaves similarly for the three different satellites in terms of SI and CORR, a substantially more (> 10%) negative model BIAS associated with a considerably lower SLOPE is found for Cryosat-2. RMSE is also higher for this satellite. Figure 6 shows that these results are largely consistent between the different Mediterranean sub-regions although they are more pronounced in the Western Mediterranean and the Adriatic Sea. A lower model underestimate of the Cryosat-2 measurements is observed in the Ionian Sea and the Eastern
Mediterranean. The statistics of model-Jason-2 and model-Saral comparisons are comparable, with the model exhibiting its best performance when compared to the observations of Saral.

It was decided to exclude the observations of Cryosat-2 from the analysis. Apart from the aforementioned discrepancies, there are other results in the literature to support this decision. Specifically, satellite-buoy comparisons performed by Sepulveda et al. (2015) have shown that Saral Hs is of better quality than Jason-2 and Cryosat-2 Hs at both
open ocean and coastal buoy sites. In fact, Saral data are of very high quality with no need of corrections whilst corrections are applied to Jason-2 and Cryosat-2 Hs observations (corrected data are used herein). After corrections, Jason-2 Hs has been found to well approximate Saral Hs whilst less accurate results have been obtained for Cryosat-2, particularly for wave heights below 1.5 m. For these reasons, in what follows, the comparison of Med-waves Hs is performed against merged satellite observations from Saral and Jason-2 satellites which are of similar accuracy.

Table 4 shows statistics from the comparison of the Med-waves hindcast Hs and satellite observations of Hs, for the full Mediterranean Sea, for 1-year period and seasonally. Figure 7 (right) shows the corresponding QQ-Scatter plot for 1-year period, for the full Mediterranean Sea. Figure 7 (left) shows an equivalent QQ-Scatter plot resulting from the comparison of the ECMWF forcing wind speeds, U10, and Jason-2 measurements of U10 (no U10 available from Saral or Cryosat-2).

Figure 7 (left) shows that the ECMWF forcing wind model underestimates observed U10 throughout the entire U10 range, even more so at high wind speeds. An overall model underestimation of 9% associated with a SLOPE of 0.9 have been computed. Figure 7 (right) also shows an overall Med-waves model underestimation of observed Hs by about 5% associated with a SLOPE of 0.96. Nevertheless, in this case, the model somewhat underestimates observed Hs over the lower Hs range (< 2 m), converges to the observed Hs over the middle Hs range (2-3.5 m) while, generally, somewhat
overestimates the larger waves in the data records. This apparent discrepancy between wind and wave scatter distributions is a consequence of the modification of the default values of the whitecapping dissipation coefficients in WAM as described in Section 3. A QQ-Scatter obtained before this modification (not shown) is indeed very similar to the one of the ECMWF wind speeds in Fig. 7. On the whole, Fig. 7 shows that the performance of Med-waves at offshore locations in the



Mediterranean Sea (satellite records near the coast are mostly filtered out as unreliable) is very good. Comparing to the equivalent results obtained from the model-buoy comparison (Fig. 4), a very similar pattern of scatter distribution is observed in the two plots, also evident from the orientation of the best-fit lines and the curvature of the QQ-plots. A smaller scatter (by about 6%) with a larger overall bias (by about 2%) is associated with the model-satellite comparison. SI values

compare well at the more exposed wave buoys in the Mediterranean Sea.

        Table 4 shows the seasonal variation of the Med-waves model performance. RMSE varies from 0.17 m in summer to 0.24 m in winter. SI is highest in summer (0.2) and lowest in winter (0.17). Correlation coefficient varies accordingly. In general, as explained in the previous sub-section, a lower scatter with a higher correlation is expected the more well-defined the weather conditions are. Alike in the model-buoy comparison, BIAS is negative in all seasons. Its highest relative value

(BIAS/$\overline{R}$) of 7.7% is computed for autumn and its lowest of 3.5% for summer. SLOPE varies from 0.95 in spring and autumn to 0.97 in winter. Overall, Table 4, alike Table 1, reveals that the statistics of spring are the most representative of the year-long statistics for the Mediterranean Sea.

        Table 5 shows the statistics of the comparison of the Med-waves hindcast Hs and satellite observations of Hs for the different sub-regions of the Mediterranean Sea defined in Fig. 3. For visualization purposes, Fig. 8 (right column) maps the

statistics shown in Table 5. In addition, equivalent statistics are mapped for the ECMWF - satellite comparison of wind speeds (left column). It is noted that the relative BIAS (BIAS/$\overline{R}$) is displayed in the figure. This quantity allows for a more straightforward comparison between the different sub-basins in terms of percentage deviations from the observed mean value. It is also highlighted that the spatial coverage of the model-satellite wind collocations (measurements only from Jason-2) is much more limited than the spatial coverage of the model-satellite wave collocations (measurements from both

Saral and Jason-2). As a consequence, the wave statistics are expected to be more representative of the sub-regions under consideration compared to the wind statistics. This is particularly true for the Adriatic, the Ligurian and the Alboran Seas. In addition, the wave statistics have been computed using a sample size of at least 400 data points whilst the wind statistics have been obtained with a minimum sample requirement of 200 data points. Thus, the confidence associated with the wave statistics is higher than the confidence associated with the wind statistics. For the above reasons, the wind metrics presented

in Fig. 8 are interpreted with caution.

        Figure 8 (right column) shows that the typical error (RMSE) varies from 0.18 m in the South-Central Levantine Basin (lev3 in Fig. 3) to 0.24 m in the Alboran Sea, the North West Mediterranean (nwm) and North Adriatic sub-regions up to 0.29 m in the Atlantic sub-region. In terms of SI, the highest value (0.28) is obtained in the North Adriatic Sea followed by the Aegean Sea (0.24). The South Adriatic, Alboran, Ligurian (tyr1), and East Levantine (lev4) Seas have also relatively

high SI values (0.21-0.23). The lowest values are found over the south eastern Mediterranean Sea (0.15-0.16) and in the Atlantic sub-region (0.15).. SI and CORR have a similar pattern of variation; a notable difference being that the correlation coefficient obtains its worst value in the East Levantine and, in general, it has relatively lower values in the well-exposed regions of the Levantine Basin compared to the well-exposed regions on its west. In accordance with the above results, Ratsimandresy et al. (2008), examining model-satellite agreement over coastal locations of the western Mediterranean Sea,



found the worst correlations in the Alboran Sea and east of Corsica Island. Bertotti et al. (2013), in a comparison of high resolution wind and wave model output with satellite data over different sub-regions of the Mediterranean Sea, also found the largest scatter and lowest correlations in the Adriatic and the Aegean Seas. In agreement, Zacharioudaki et al. (2015), focusing on the Greek Seas, have shown a considerably larger scatter in the Aegean Sea than in the surrounding seas, when

model output was compared to satellite observations. As explained in the previous sub-section (model-buoy comparison), it is the difficulties of wind models to well-reproduce orographic effects and/or local sea breezes and the difficulties of wave models to well-resolve complicated bathymetry that introduce errors in these fetch-limited, enclosed regions, often characterized by a complex topography. Indeed, comparison with the equivalent results for the ECMWF wind speeds confirms these difficulties. For example, the pattern of SI and CORR variation for U10 largely resembles that for Hs,

corroborating the conclusion of many studies that errors in wave height simulations by sophisticated wave models are mainly caused by errors in the generating wind fields (e.g. Komen et al., 1994; Ardhuin et al., 2007). Nevertheless, some differences do exist. For instance, the Hs SI in the Aegean Sea is relatively higher than the corresponding U10 SI. This is most probably because in this region of highly complicated bathymetry with many little islands the error of the wave model increases in relation to the error of the wind model. Similarly, in the East Levantine, Hs SI is lower than that implied by U10 SI. In this

case, the wind model may not well simulate local wind patterns, characterized by local sea breezes and easterly directions (Galil et al., 2006), however, the wave regime which is dominated by waves from the west sector (Galil et al., 2006) is better reproduced by the wave model. Negative BIAS and SLOPE below unity are the case in all sub-regions except for the Atlantic and Alboran Sea. In the latter regions, the wave model overestimates the observations by 5-6%. Otherwise, it underestimates the observations by about 2% in the East South-West Mediterranean (swm1) and Aegean Seas to about 15%

in the Adriatic (adr2). In general, the largest biases are found in the Adriatic (adr1, adr2), the North Ionian Sea (ion3) and the Levantine Basin (lev2,lev3,lev4) with values of 7.5 - 15.2%. SLOPE varies accordingly with values between 0.88 in the South Adriatic Sea (adr2) and 0.99 in the Aegean Sea  up to 1.05 in the Alboran Sea. Comparing with the equivalent results for the ECMWF wind speed, it is evident that although there are similarities in the relative BIAS and SLOPE distributions, there are also considerable differences. In general, in terms of absolute value, the relative BIAS associated with the wind

field is larger than that associated with the wave field except for the South Adriatic Sea, the Alboran Sea and the Atlantic. In fact, in the latter two regions, a change of sign from negative to positive is observed between wind and waves. As already mentioned, this is a consequence of the modification of the whitecapping dissipation coefficients from default values in WAM, which has led to an important offset of the negative BIAS associated with the ECMWF wind speeds, especially over the high Hs range. Thus, in regions where the ECMWF underestimate has been small, as in the Atlantic, modification of the

dissipation coefficients has eventually led to an overshoot of the observed Hs. This is a robust pattern obtained for the whole Atlantic area simulated by the nested Med-waves model (up to -18.125° W, Fig. 1). The increase in negative Hs relative BIAS in the South Adriatic Sea relative to the respective U10 relative BIAS is somewhat puzzling, however, as mentioned above, small confidence is pertained to the results of U10 in the Adriatic Sea due to a limited observational coverage by Jason-2.



## 4.2    Hindcast mean wave period

Table 6 presents the statistics of the comparison between the Med-waves hindcast mean wave period, Tm, and in-situ observations of mean wave period, for the full Mediterranean Sea, for 1 year period (2014) and seasonally. Figure 9

shows the corresponding QQ-Scatter plot for the year-long statistics. It is shown that the model exhibits greater variability than the observations (STD in Table 6). RMSE varies from 0.8 s in summer to 1.07 s in winter. In relation to the mean of the observations, the error is about 17-19%, with winter and spring being at the low end of this range and autumn at the high. SI varies from 0.12 in winter and spring to 0.14 in summer and autumn. The non-trivial deviation of SI from relative RMSE (RMSE/$\overline{R}$) indicates that a substantial part of the error is caused by BIAS. CORR has its minimum value (0.78) in summer

and its maximum (0.87) in winter and spring. As before, these results indicate that the model wave period, alike the model wave height, better follows the observations in well-defined wave conditions of higher waves and larger periods. BIAS is negative with values that correspond to a model underestimate of about 11.5-13%. Correspondingly, SLOPE has a small variation of 0.87-0.89. Like for wave height, spring statistics are the most representative of the year-long statistics. Figure 9 clearly shows that the wave model underestimates the observations throughout the observed Tm range. Measurements of Tm

< 4.5 s are especially underestimated while those of relatively high Tm are better approximated by the model.

Table 7 gives the statistics of the model-buoy comparison at the individual wave buoy locations. The typical error relative to the mean of the observations (RMSE/$\overline{R}$) has its lowest values of 12-16% over the western part of the Mediterranean Sea, west and south of France, with the two locations nearest to the Gibraltar Straight being at the low end of this range. Otherwise, this error is 17-23% reaching up to 29% at location 61187 near the French-Italian border. At this

location, all qualification metrics obtain their worst value. This is because wave buoy 61187 is located at a distance below 2 km from coast and is affected by winds blowing from land. As already explained in the model-buoy wave height comparison, in this situation, the simulated fetch may differ substantially from the actual fetch because of the wave model grid size approximation; moreover, wind speed and wave height is considerably underestimated. RMSE is mainly caused by BIAS which is negative at all locations. Thus, accordingly to the RMSE, the relative BIAS is below 10% over the western

Mediterranean, is 14%-20% otherwise reaching up to 23% at location 61187. It is only at location 61197, offshore from the eastern Balearic islands, that the scatter of the data appears to contribute more to the typical error than the bias. This is a well-exposed offshore location where BIAS (2%) and SLOPE (0.97) have their best values and where model performance has been found to be optimal for wave height. The relatively high SI (0.15) and moderate correlation (0.86) at this location could be associated with the appearance of two density peaks in the density Scatter plot (not shown), indicative of a double

peaked frequency spectrum. Density scatter plots with two peaks, although less distinct, have also been obtained for locations 61289 and 61021, offshore from France. In general, a close examination of the QQ-Scatter plots (not shown) corresponding to the different locations has revealed that the model largely underestimates the observed Tm over the lower wave period range at all locations. Over the higher range, the model converges or overestimates the observed Tm in the





western Mediterranean Sea, west and south of France. Otherwise, model underestimates all observed Tm with some convergence towards higher values. SLOPE mostly follows the pattern of variation of BIAS with values between 0.76 and 0.97. SI is relatively small with values between 0.09 (ATHOS) and 0.18 (61187) while CORR varies from 0.65 (61187) to 0.9 (61430, 68422). Generally, similarly to the wave height results, the lowest correlations are found at coastal locations affected by fetch differences between model and reality due to a complex surrounding topography. On the other hand, the highest correlations are obtained at the most exposed wave buoy locations.

### 4.3    Forecast skill

In the previous section, the performance of the Med-waves system has been characterized through the comparison of hindcast wave parameters with observations. In this section, the forecast skill of the Med-waves system is explored by comparing forecast wave parameters with observations at different forecast lead-times. Hence, Fig. 10 shows Med-waves forecast skill for Hs (bottom two rows) together with ECMWF forecast skill for U10 (top row). The latter is evaluated against satellite observations, the former is evaluated against satellite (middle row) and buoy (bottom row) observations. It is noted that in the model-buoy comparisons, each lead-time represents a single point in time, whist, in the model-satellite comparisons, each lead-time represents a full forecast day, i.e. +0 h represents forecast day 1 containing data from 0 h to 24 h forecast. This approach is dictated by the scarcity of satellite observations in time.

Figure 10 shows that Hs SI grows with forecast lead-time at a constantly increasing growth rate. At the same time, CORR decreases with forecast lead-time with the decrease being more notable after the 3$^{rd}$ day of forecast (+48 h). These patterns, which are consistent between model-buoy and model-satellite observations, mostly agree with the equivalent U10 patterns and manifest the deterioration of the forecast in time. A small difference between U10 and Hs forecast skill is the somewhat more linear increase of SI with forecast lead-time in the first case which results in a smaller overall U10 deterioration over the length of the forecast (13%) compared to the respective Hs deterioration (19% for model-satellite comparisons). This is indicative of the sensitivity of wave height to even limited variations of the input wind intensity. On the other hand, waves seem to be less sensitive to wind misfits in time and space which is manifested by the higher and more persistent Hs CORR over the forecast range compared to the respective U10 CORR. Contrary to SI and CORR but also to U10 BIAS, the evolution of Hs BIAS with forecast lead-time in not monotonic. This apparent discrepancy between wind and wave BIAS evolution is attributed to the modification of the default values of the whitecapping dissipation coefficients in WAM, which, as shown in Section 4.1.2, have an impact on the bias of the wave model output relative to the observations. In any case, for both winds and waves, the variation of BIAS with forecast range is small and does not exceed 3%.

Figure 11, alike Fig. 10 (bottom row), shows Med-waves forecast skill for Tm evaluated against wave buoy observations. Similarly to Hs, SI increases with forecast range, CORR decreases and BIAS exhibits a non-monotonic variation analogous to the one of Hs. In this case however, the variation of SI over the forecast period is small (5%) compared to the respective Hs variation (25%). This agrees with the finding that Tm errors are mainly caused by BIAS (Section 4.2).




## 5      Conclusions

The CMEMS Mediterranean wave forecasting system, Med-waves, is operational since April 2016 providing short-range forecasts over the Mediterranean Sea at hourly intervals and at an horizontal resolution of 1/24°. The development and the evaluation of the performance of the system has been presented in detail in this paper. In the framework of this

evaluation, the wave parameters of significant wave height and mean wave period have been evaluated against in-situ and satellite observations over a period of one year (2014). Both hindcast quality and forecast skill have been assessed. In the former case, evaluation statistics have been provided for the Mediterranean Sea as a whole, at individual buoy locations and over pre-defined Mediterranean sub-regions. In the latter case, evaluation statistics have been provided only for the entire Mediterranean Sea. The main findings of this evaluation assessment are summarized below.

Overall, the significant wave height is accurately simulated by the model. Considering the Mediterranean Sea as a whole, the RMSE is 0.21 m and the bias is -0.03 m (3.7%) when the model is compared to in-situ observations and -0.06 m (5.5%) when it is compared to satellite observations. In general, the model somewhat underestimates the observations for wave heights below 4 m whilst it mostly converges to the observations for higher waves. The scatter index, indicative of the scatter of the data around their regression line, is 25% for the model-in-situ comparison and 19% for the model-satellite

comparison demonstrating a reduced scatter offshore (where satellite measurements are mostly located) compared to nearshore (where in-situ measurements are mostly obtained). The correlation coefficient is 0.95-0.96 and so is the best-fit slope. Model performance is better in winter when the wave conditions are well-defined. Spatially, the model performs optimally at offshore wave buoy locations and well-exposed Mediterranean sub-regions. Within enclosed basins and near the coast, unresolved topography by the wind and wave models and fetch limitations cause the wave model performance to

deteriorate. In particular, the model has an optimal performance along most of the southern Mediterranean Sea. Its performance is less good in the Alboran, Ligurian, Adriatic, Aegean and Eastern Levantine Seas, with the worst evaluation statistics obtained in the Adriatic. In terms of bias, the model overall underestimates the measurements in the Mediterranean Sea. The smallest underestimate is observed in the Aegean Sea while overestimate is observed in the Alboran Sea and in the Atlantic.

The mean wave period is reasonably well simulated by the model. The RMSE is 0.7 s and is mainly caused by model bias which has a value of -0.48 s (12%). In general, the model underestimates the observed mean wave period and exhibits greater variability than the observations. A relatively larger model underestimate is found for mean wave periods below 4.5 s. The scatter index is 13%, the correlation coefficient is 0.85 and the best-fit slope is 0.88. Model performance is a little better in winter when wave conditions are well-defined. Spatially, the model somewhat overestimates the highest

mean wave period values in the western Mediterranean Sea, west and south of France. Otherwise, model underestimate is widespread. Similarly to the wave height, the model performance is best at well-exposed offshore locations and deteriorates near the shore mainly due to fetch limitations.



The forecast skill of the model over the Mediterranean Sea deteriorates with forecast range. The growth of error in the wave forecast is mainly due to the growth of error in the forcing wind fields. The scatter index of the significant wave height deteriorates by 19% and 25% over the 5-day forecast for model-satellite and model-buoy comparisons respectively. The equivalent deterioration for mean wave period is only 5% (model-buoy comparison). A monotonic decrease in

correlation is also observed. On the contrary, the evolution of bias with forecast range shows some variability with no clear trend. Nevertheless, this variability does not exceed 3% over the forecast period.

In the near future an Optimal Interpolation type data assimilation scheme will be added to the Med-waves system in order to blend satellite along-track significant wave height measurements with model background forecasts. Although wave data assimilation is known not to be particularly beneficial in areas where wind sea conditions are dominant we expect that

wave forecasts in certain sub-areas of the Mediterranean Sea where swell propagation is quite frequent, will be improved at +24h and perhaps +48h lead time. The enhanced Med-waves system with the data assimilation system module is going to produce 3-hourly wave analyses on a daily basis for the Mediterranean Sea by assimilating Sentinel-3 and Jason-3 altimeter measured significant wave heights and surface winds. The assimilation will be based on the inherent data assimilation scheme of WAM Cycle 4.5.4 model which generates an updated wave field by distributing the information from the

observed significant wave height and surface wind data within a given time window over the entire model grid. The Med-waves Data Assimilation component is planned to be added to the Med-waves system by April 2018.

Another planned improvement is the future usage of higher than 1/8° horizontal resolution wind analyses and forecasts to force the Med-waves system with expected potential impacts to near coastal areas of the Mediterranean Sea already well resolved by the high resolution grid of  the wave model.

**Data availability**

The in-situ wave buoy observations used in this study have been obtained from the Copernicus Marine Environment Monitoring Service (CMEMS) IN-SITU Thematic Assembly Centre (INS TAC) archive and are available from http://marine.copernicus.eu/services-portfolio/access-to-products/?option=com_csw&view=details&product_id=INSITU_MED_NRT_OBSERVATIONS_013_035.    The    satellite
observations have been obtained from a merged altimeter wave height database setup at CERSAT - IFREMER, France, and are available from ftp://ftp.ifremer.fr/ifremer/cersat/products/swath/altimeters/waves/data/. The model outputs for year 2014 are available upon request from the authors. Model outputs since year 2015 are available through CMEMS from http://marine.copernicus.eu/services-portfolio/access-to-products/?option=com_csw&view=details&product_id=MEDSEA_ANALYSIS_FORECAST_WAV_006_01.





**Competing interests**

The authors declare that they have no conflict of interest.

**Acknowledgements**

This work has been supported by HCMR (www.hcmr.gr/) and Copernicus Marine Environment Monitoring Service
5   (CMEMS) (http://marine.copernicus.eu/). CMEMS is implemented by Mercator Ocean through a Delegation Agreement
with European Union. The authors want to acknowledge the CMEMS for providing ocean currents data and in-situ
observations, the Italian Meteorological Service for providing ECMWF wind data and CERSAT – IFREMER for the
satellite observations for validating the system.




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





**Table 1: Med-waves Hs evaluation against wave buoys' Hs, for the full Mediterranean Sea, for 1 year period (2014) and seasonally.**

| MED | ENTRIES | $\overline{R}$ (m) | $\overline{M}$ (m) | STD R (m) | STD M (m) | RMSE (m) | SI | BIAS (m) | CORR | SLOPE |
|---|---|---|---|---|---|---|---|---|---|---|
| Whole Year | 67804 | 0.81 | 0.78 | 0.63 | 0.63 | 0.21 | 0.25 | -0.03 | 0.95 | 0.95 |
| Winter | 14136 | 1.06 | 1.02 | 0.78 | 0.77 | 0.25 | 0.24 | -0.04 | 0.95 | 0.95 |
| Spring | 19350 | 0.81 | 0.79 | 0.64 | 0.62 | 0.2 | 0.25 | -0.02 | 0.95 | 0.95 |
| Summer | 18860 | 0.64 | 0.62 | 0.45 | 0.45 | 0.17 | 0.26 | -0.02 | 0.93 | 0.96 |
| Autumn | 15458 | 0.78 | 0.74 | 0.58 | 0.6 | 0.21 | 0.27 | -0.04 | 0.94 | 0.96 |





**Table 2: Med-waves Hs evaluation against wave buoys' Hs, for each individual buoy location, for 1 year period (2014).**

| Buoy ID | ENTRIES | $\overline{R}$ (m) | $\overline{M}$ (m) | STD R (m) | STD M (m) | RMSE (m) | SI | BIAS (m) | CORR | SLOPE |
|---|---|---|---|---|---|---|---|---|---|---|
| 61198 | 2464 | 0.97 | 1.04 | 0.66 | 0.69 | 0.21 | 0.21 | 0.07 | 0.96 | 1.05 |
| 61417 | 2746 | 1.02 | 0.97 | 0.58 | 0.55 | 0.20 | 0.19 | -0.05 | 0.94 | 0.93 |
| 61281 | 2140 | 0.78 | 0.75 | 0.4 | 0.39 | 0.19 | 0.24 | -0.03 | 0.89 | 0.94 |
| 61280 | 2324 | 0.79 | 0.73 | 0.46 | 0.45 | 0.19 | 0.23 | -0.06 | 0.92 | 0.92 |
| 61430 | 2520 | 0.95 | 0.92 | 0.66 | 0.64 | 0.22 | 0.23 | -0.03 | 0.94 | 0.96 |
| 61197 | 2767 | 1.30 | 1.27 | 0.92 | 0.89 | 0.23 | 0.17 | -0.04 | 0.97 | 0.96 |
| 61196 | 2884 | 1.24 | 1.26 | 0.88 | 0.91 | 0.27 | 0.22 | 0.02 | 0.95 | 1.00 |
| 61188 | 2324 | 0.64 | 0.59 | 0.50 | 0.47 | 0.19 | 0.28 | -0.05 | 0.93 | 0.90 |
| 61191 | 2289 | 0.66 | 0.65 | 0.52 | 0.52 | 0.16 | 0.25 | -0.01 | 0.95 | 0.97 |
| 61190 | 2106 | 0.63 | 0.67 | 0.54 | 0.54 | 0.18 | 0.28 | 0.04 | 0.95 | 1.01 |
| 61431 | 654 | 0.81 | 0.79 | 0.55 | 0.49 | 0.18 | 0.22 | -0.02 | 0.95 | 0.93 |
| 61289 | 2261 | 0.92 | 0.91 | 0.58 | 0.58 | 0.17 | 0.18 | -0.01 | 0.96 | 0.98 |
| 61021 | 1561 | 1.07 | 1.18 | 0.64 | 0.79 | 0.35 | 0.32 | 0.11 | 0.91 | 1.10 |
| 61187 | 1496 | 0.53 | 0.50 | 0.35 | 0.32 | 0.17 | 0.30 | -0.04 | 0.89 | 0.89 |
| 61213 | 2024 | 1.10 | 1.04 | 0.99 | 0.92 | 0.23 | 0.20 | -0.07 | 0.98 | 0.93 |
| 61295 | 871 | 0.57 | 0.49 | 0.39 | 0.37 | 0.16 | 0.25 | -0.07 | 0.93 | 0.87 |
| 61221 | 2716 | 0.63 | 0.70 | 0.41 | 0.46 | 0.19 | 0.27 | 0.07 | 0.93 | 1.09 |
| 61219 | 2471 | 0.80 | 0.70 | 0.56 | 0.51 | 0.19 | 0.21 | -0.09 | 0.96 | 0.88 |
| 61216 | 2347 | 0.64 | 0.59 | 0.47 | 0.43 | 0.16 | 0.24 | -0.05 | 0.95 | 0.90 |
| 61214 | 2705 | 0.87 | 0.81 | 0.68 | 0.63 | 0.19 | 0.21 | -0.06 | 0.96 | 0.92 |
| 61211 | 2705 | 0.66 | 0.59 | 0.60 | 0.53 | 0.20 | 0.28 | -0.07 | 0.95 | 0.87 |
| 61209 | 1141 | 0.77 | 0.73 | 0.62 | 0.60 | 0.19 | 0.24 | -0.04 | 0.96 | 0.94 |
| 61208 | 1708 | 0.78 | 0.75 | 0.55 | 0.50 | 0.16 | 0.21 | -0.03 | 0.96 | 0.93 |
| 61207 | 2185 | 0.57 | 0.49 | 0.47 | 0.44 | 0.17 | 0.27 | -0.08 | 0.95 | 0.87 |
| 61210 | 2488 | 0.68 | 0.63 | 0.54 | 0.52 | 0.18 | 0.26 | -0.05 | 0.95 | 0.92 |
| 61215 | 2712 | 0.65 | 0.55 | 0.46 | 0.44 | 0.20 | 0.26 | -0.10 | 0.93 | 0.86 |
| 61218 | 631 | 0.80 | 0.68 | 0.69 | 0.59 | 0.44 | 0.53 | -0.13 | 0.79 | 0.77 |
| 61220 | 2411 | 0.50 | 0.50 | 0.44 | 0.44 | 0.21 | 0.43 | 0.00 | 0.88 | 0.96 |
| 68422 | 1862 | 0.99 | 0.94 | 0.67 | 0.60 | 0.22 | 0.22 | -0.05 | 0.95 | 0.92 |
| 61277 | 1945 | 1.01 | 0.99 | 0.59 | 0.59 | 0.22 | 0.21 | -0.02 | 0.93 | 0.97 |
| SARON | 2770 | 0.45 | 0.46 | 0.26 | 0.35 | 0.18 | 0.40 | 0.01 | 0.87 | 1.05 |
| ATHOS | 1576 | 0.82 | 0.81 | 0.60 | 0.61 | 0.22 | 0.27 | -0.01 | 0.94 | 0.98 |





**Table 3: Med-waves Hs evaluation against satellite Hs, for the full Mediterranean Sea, for 1 year period (2014).**

| Satellite | ENTRIES | $\bar{R}$ (m) | $\bar{M}$ (m) | STD R (m) | STD M (m) | RMSE (m) | SI | BIAS (m) | CORR | SLOPE |
|---|---|---|---|---|---|---|---|---|---|---|
| Jason-2 | 14268 | 1.15 | 1.08 | 0.74 | 0.76 | 0.23 | 0.19 | -0.07 | 0.96 | 0.95 |
| Saral | 14877 | 1.04 | 1.00 | 0.74 | 0.75 | 0.20 | 0.18 | -0.04 | 0.97 | 0.97 |
| Cryosat-2 | 13939 | 1.21 | 1.01 | 0.69 | 0.75 | 0.3 | 0.18 | -0.20 | 0.96 | 0.88 |

**Table 4: Med-waves Hs evaluation against satellite Hs (Jason-2 and Saral), for the full Mediterranean Sea, for 1 year period (2014) and seasonally.**

| MED | ENTRIES | $\bar{R}$ (m) | $\bar{M}$ (m) | STD R (m) | STD M (m) | RMSE (m) | SI | BIAS (m) | CORR | SLOPE |
|---|---|---|---|---|---|---|---|---|---|---|
| Whole Year | 29145 | 1.09 | 1.04 | 0.74 | 0.75 | 0.21 | 0.19 | -0.06 | 0.96 | 0.96 |
| Winter | 7316 | 1.36 | 1.31 | 0.91 | 0.94 | 0.24 | 0.17 | -0.06 | 0.97 | 0.97 |
| Spring | 7341 | 1.10 | 1.04 | 0.73 | 0.72 | 0.21 | 0.19 | -0.06 | 0.96 | 0.95 |
| Summer | 7328 | 0.86 | 0.83 | 0.47 | 0.48 | 0.17 | 0.2 | -0.03 | 0.94 | 0.96 |
| Autumn | 7160 | 1.04 | 0.97 | 0.71 | 0.72 | 0.21 | 0.19 | -0.08 | 0.96 | 0.95 |

**Table 5: Med-waves Hs evaluation against satellite Hs (Jason-2 and Saral), for each individual Mediterranean Sea sub-region shown in Fig. 3, for 1 year period (2014).**

| Satellite | ENTRIES | $\bar{R}$ (m) | $\bar{M}$ (m) | STD R (m) | STD M (m) | RMSE (m) | SI | BIAS (m) | CORR | SLOPE |
|---|---|---|---|---|---|---|---|---|---|---|
| atl | 1216 | 1.79 | 1.89 | 1.02 | 1.00 | 0.29 | 0.15 | 0.10 | 0.96 | 1.03 |
| alb | 681 | 1.03 | 1.08 | 0.67 | 0.74 | 0.24 | 0.23 | 0.05 | 0.95 | 1.05 |
| swm1 | 2024 | 1.18 | 1.16 | 0.74 | 0.76 | 0.21 | 0.18 | -0.02 | 0.96 | 0.98 |
| swm2 | 1296 | 1.29 | 1.24 | 0.96 | 0.97 | 0.23 | 0.17 | -0.05 | 0.97 | 0.97 |
| nwm | 3413 | 1.31 | 1.25 | 0.95 | 0.96 | 0.24 | 0.18 | -0.06 | 0.97 | 0.96 |
| tyr1 | 554 | 0.89 | 0.84 | 0.59 | 0.62 | 0.20 | 0.22 | -0.05 | 0.95 | 0.96 |
| tyr2 | 2679 | 1.09 | 1.04 | 0.75 | 0.78 | 0.22 | 0.19 | -0.05 | 0.96 | 0.97 |
| ion1 | 1894 | 1.13 | 1.09 | 0.79 | 0.79 | 0.20 | 0.18 | -0.04 | 0.97 | 0.97 |
| ion2 | 4598 | 1.16 | 1.11 | 0.78 | 0.78 | 0.20 | 0.16 | -0.05 | 0.97 | 0.96 |
| ion3 | 1661 | 1.03 | 0.93 | 0.72 | 0.69 | 0.21 | 0.18 | -0.10 | 0.97 | 0.91 |
| adr1 | 809 | 0.80 | 0.72 | 0.60 | 0.66 | 0.24 | 0.28 | -0.08 | 0.94 | 0.95 |
| adr2 | 655 | 0.89 | 0.75 | 0.54 | 0.54 | 0.23 | 0.22 | -0.13 | 0.94 | 0.88 |
| lev1 | 1572 | 1.15 | 1.11 | 0.63 | 0.64 | 0.19 | 0.16 | -0.05 | 0.96 | 0.96 |
| lev2 | 2236 | 1.04 | 0.96 | 0.58 | 0.57 | 0.20 | 0.18 | -0.08 | 0.95 | 0.93 |
| lev3 | 1606 | 1.04 | 0.96 | 0.56 | 0.54 | 0.18 | 0.15 | -0.08 | 0.96 | 0.93 |
| lev4 | 1496 | 0.84 | 0.76 | 0.42 | 0.40 | 0.19 | 0.21 | -0.08 | 0.91 | 0.89 |
| aeg | 1982 | 0.87 | 0.85 | 0.59 | 0.63 | 0.22 | 0.25 | -0.02 | 0.94 | 0.99 |



**Table 6: Med-waves Tm evaluation against wave buoys' Hs, for the full Mediterranean Sea, for 1 year period (2014) and seasonally.**

| MED | ENTRIES | $\overline{R}$ (s) | $\overline{M}$ (s) | STD R (s) | STD M (s) | RMSE (s) | SI | BIAS (s) | CORR | SLOPE |
|---|---|---|---|---|---|---|---|---|---|---|
| Whole Year | 37247 | 3.9 | 3.42 | 0.88 | 0.97 | 0.7 | 0.13 | -0.48 | 0.85 | 0.88 |
| Winter | 7987 | 4.24 | 3.73 | 0.98 | 1.07 | 0.73 | 0.12 | -0.51 | 0.87 | 0.88 |
| Spring | 11170 | 3.93 | 3.44 | 0.88 | 0.98 | 0.68 | 0.12 | -0.48 | 0.87 | 0.88 |
| Summer | 9665 | 3.57 | 3.16 | 0.66 | 0.80 | 0.65 | 0.14 | -0.41 | 0.78 | 0.89 |
| Autumn | 8425 | 3.92 | 3.41 | 0.86 | 0.94 | 0.75 | 0.14 | -0.51 | 0.82 | 0.87 |

**Table 7: Med-waves Tm evaluation against wave buoys' Tm, for each individual buoy location, for 1 year period (2014).**

| Buoy ID | ENTRIES | $\overline{R}$ (m) | $\overline{M}$ (m) | STD R (m) | STD M (m) | RMSE (m) | SI | BIAS (m) | CORR | SLOPE |
|---|---|---|---|---|---|---|---|---|---|---|
| 61198 | 2447 | 3.76 | 3.56 | 0.76 | 0.89 | 0.47 | 0.11 | -0.20 | 0.88 | 0.95 |
| 61417 | 2740 | 3.97 | 3.72 | 0.75 | 0.86 | 0.47 | 0.10 | -0.25 | 0.89 | 0.94 |
| 61281 | 2125 | 3.59 | 3.28 | 0.65 | 0.78 | 0.53 | 0.12 | -0.31 | 0.84 | 0.92 |
| 61280 | 2306 | 3.71 | 3.30 | 0.66 | 0.74 | 0.58 | 0.11 | -0.41 | 0.84 | 0.89 |
| 61430 | 2500 | 4.20 | 3.71 | 0.87 | 1.06 | 0.68 | 0.11 | -0.50 | 0.90 | 0.89 |
| 61197 | 2748 | 4.28 | 4.19 | 1.19 | 1.21 | 0.63 | 0.15 | -0.09 | 0.86 | 0.97 |
| 61196 | 2890 | 4.41 | 3.79 | 0.79 | 0.92 | 0.75 | 0.10 | -0.62 | 0.88 | 0.86 |
| 61188 | 2028 | 3.53 | 3.03 | 0.75 | 0.77 | 0.69 | 0.13 | -0.50 | 0.81 | 0.86 |
| 61191 | 1800 | 3.45 | 2.92 | 0.80 | 0.85 | 0.67 | 0.12 | -0.53 | 0.88 | 0.85 |
| 61190 | 1773 | 3.42 | 2.92 | 0.87 | 0.87 | 0.67 | 0.13 | -0.50 | 0.88 | 0.85 |
| 61431 | 621 | 3.82 | 3.10 | 0.78 | 0.84 | 0.88 | 0.13 | -0.72 | 0.81 | 0.81 |
| 61289 | 2157 | 3.82 | 3.30 | 0.70 | 0.78 | 0.66 | 0.11 | -0.52 | 0.85 | 0.87 |
| 61021 | 1531 | 4.32 | 3.77 | 0.90 | 0.98 | 0.76 | 0.12 | -0.55 | 0.85 | 0.88 |
| 61187 | 1410 | 4.33 | 3.33 | 1.00 | 0.80 | 1.26 | 0.18 | -1.00 | 0.65 | 0.76 |
| 61295 | 802 | 3.66 | 2.97 | 0.75 | 0.79 | 0.82 | 0.12 | -0.69 | 0.83 | 0.81 |
| 68422 | 2008 | 4.26 | 3.55 | 0.87 | 1.07 | 0.85 | 0.11 | -0.71 | 0.90 | 0.84 |
| 61277 | 2112 | 4.02 | 3.45 | 0.68 | 0.81 | 0.70 | 0.10 | -0.56 | 0.85 | 0.86 |
| SARON | 1809 | 3.18 | 2.75 | 0.45 | 0.60 | 0.60 | 0.13 | -0.43 | 0.72 | 0.87 |
| ATHOS | 1440 | 3.86 | 3.10 | 0.70 | 0.77 | 0.83 | 0.09 | -0.76 | 0.89 | 0.81 |




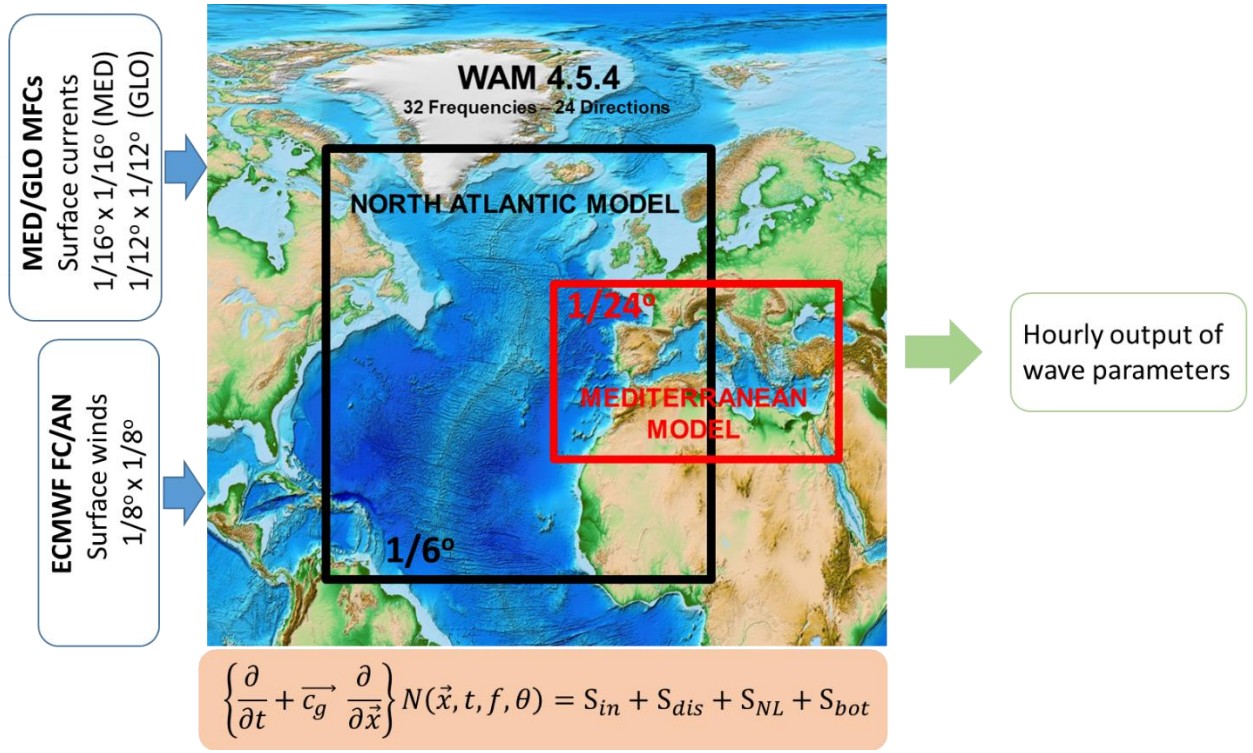

**Figure 1: Schematic of the Med-waves system.**

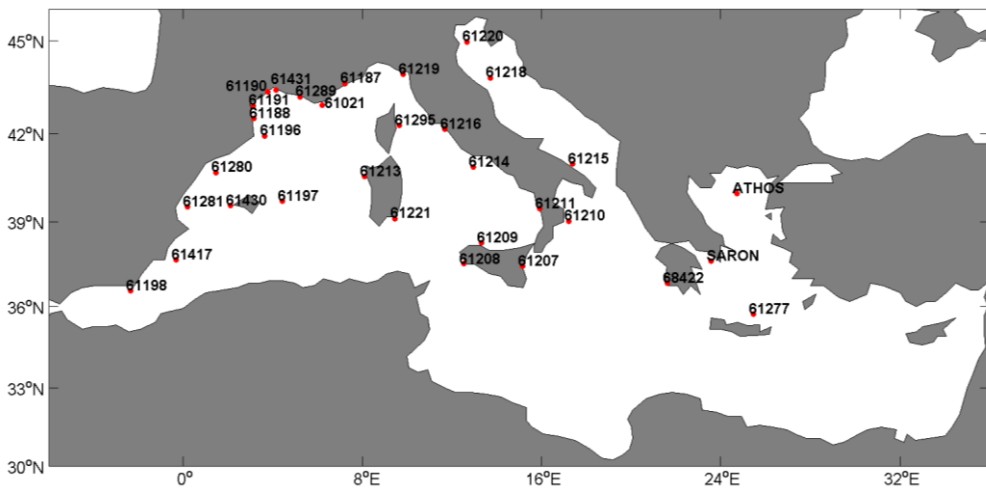

**Figure 2: Wave buoys' location and unique ID code.**





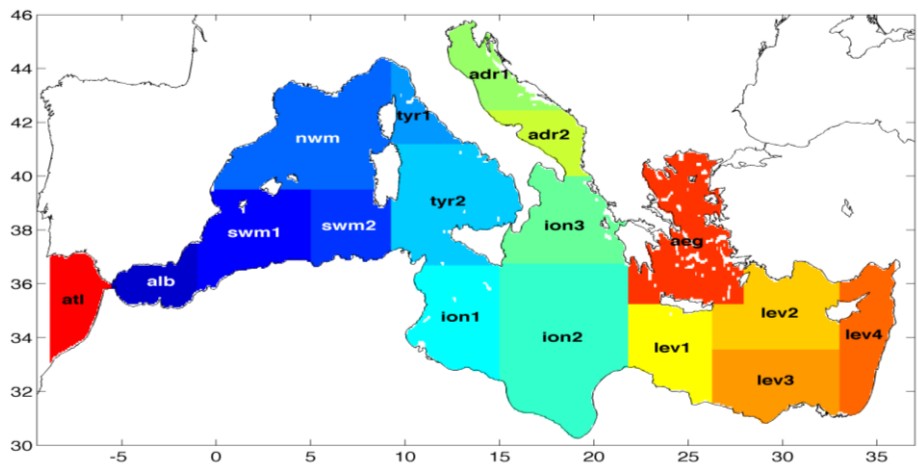

**Figure 3: Mediterranean Sea sub-regions for qualification metrics.**

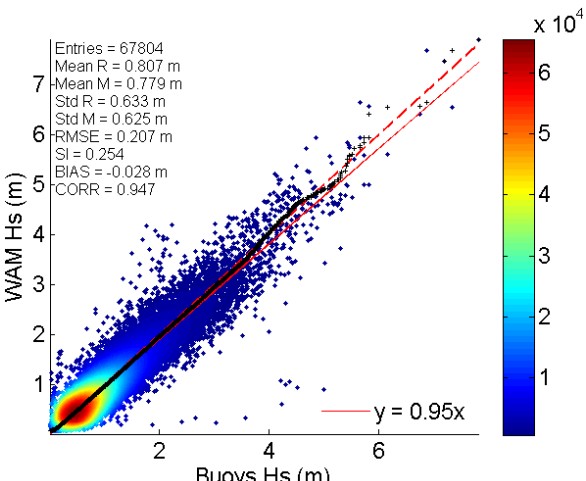

**Figure 4: QQ-Scatter plots of Med-waves output Hs versus wave buoys' observations, for the full Mediterranean Sea, for 1 year**
5 **period (2014): QQ-plot (black crosses), 45° reference line (dashed red line), least-squares best fit line (red line).**




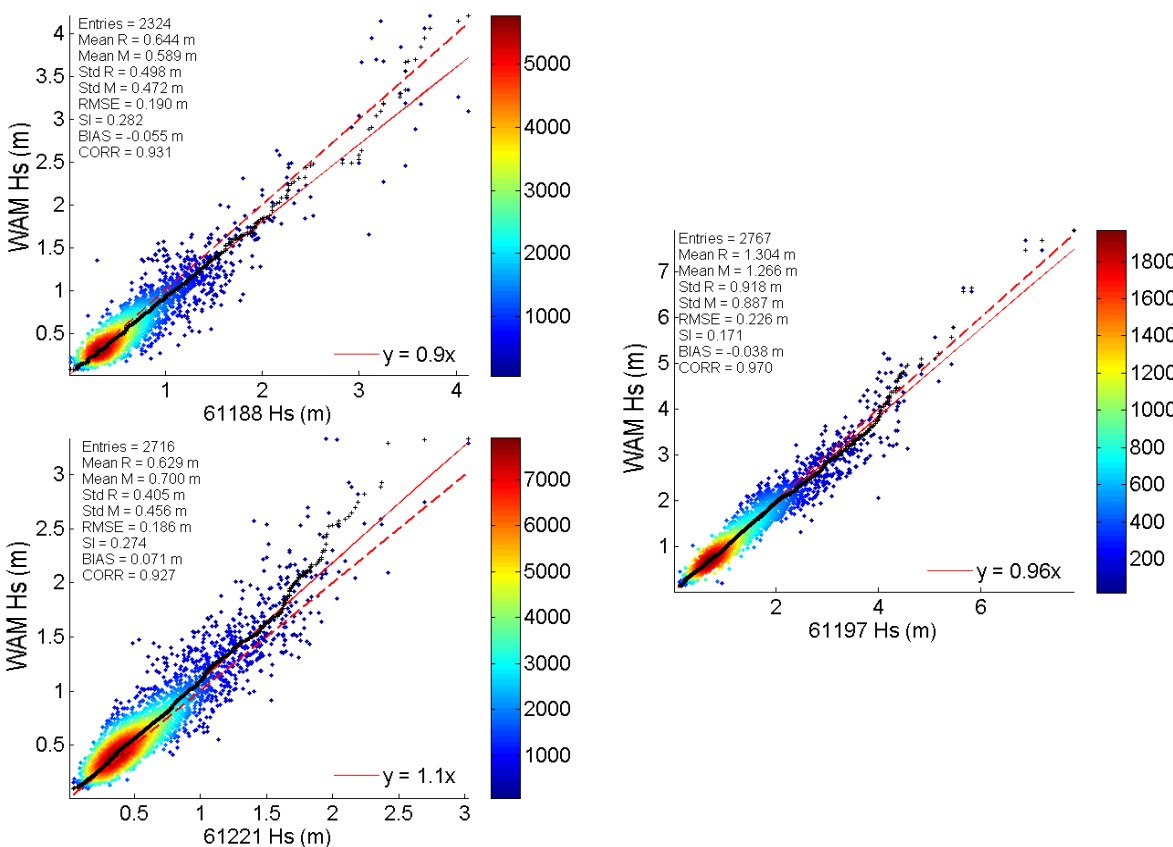

**Figure 5: QQ-Scatter plots of Med-waves output Hs versus wave buoy observations at specific wave buoy locations, for 1 year period (2014): QQ-plot (black crosses), 45° reference line (dashed red line), least-squares best fit line (red line).**





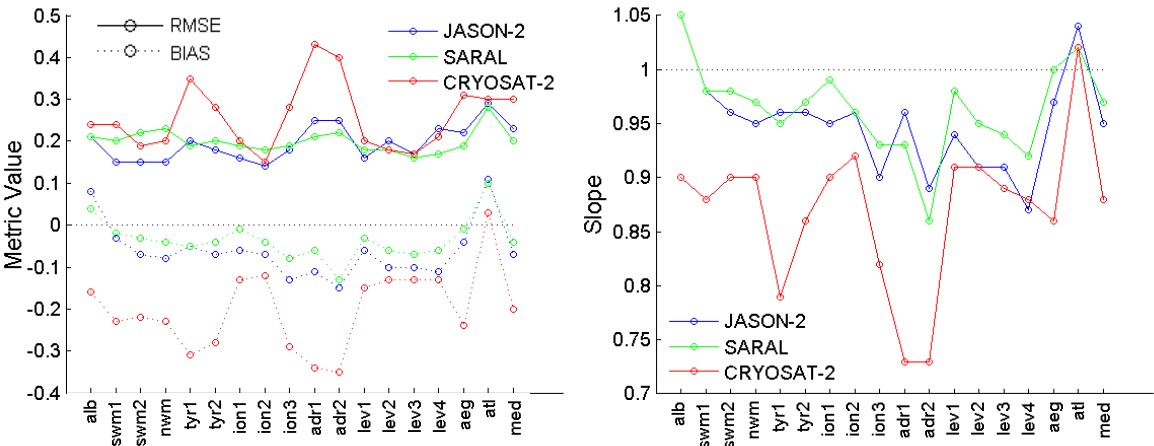

**Figure 6: Med-waves Hs evaluation against satellite Hs, for each Mediterranean Sea sub-region shown in Fig. 3, for 1 year period (2014).**

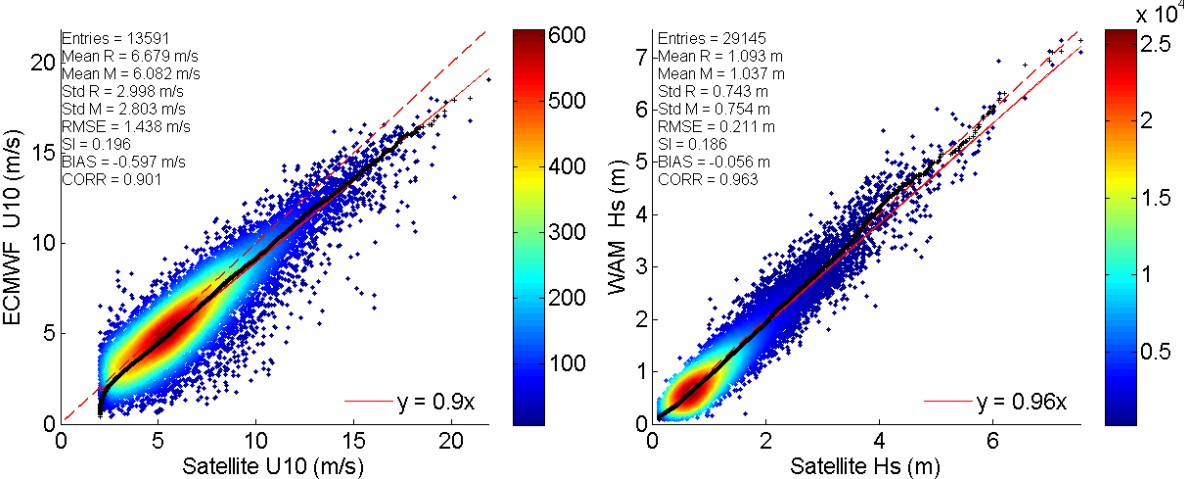

5   **Figure 7: QQ-Scatter plots of: (left) ECMWF forcing wind speed U10 versus satellite U10 (Jason-2); (right) Med-waves Hs versus satellite Hs (Jason-2 and Saral), for the full Mediterranean Sea, for 1 year period (2014).**







**Figure 8: ECMWF U10 (left column) and Med-waves Hs (right column) evaluation against satellite U10 (Jason-2) and satellite Hs (Jason-2 and Saral) respectively: maps of metric values over the Mediterranean Sea sub-regions shown in Fig. 3, for 1 year period (2014).**





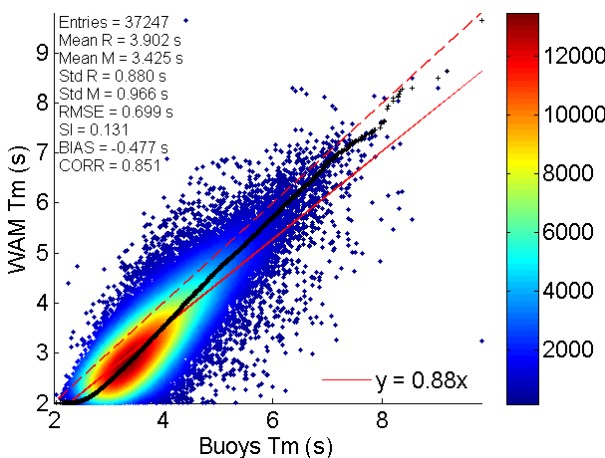

**Figure 9: QQ-Scatter plots of Med-waves output versus wave buoys' observations, for the full Mediterranean Sea, for 1 year period (2014).**





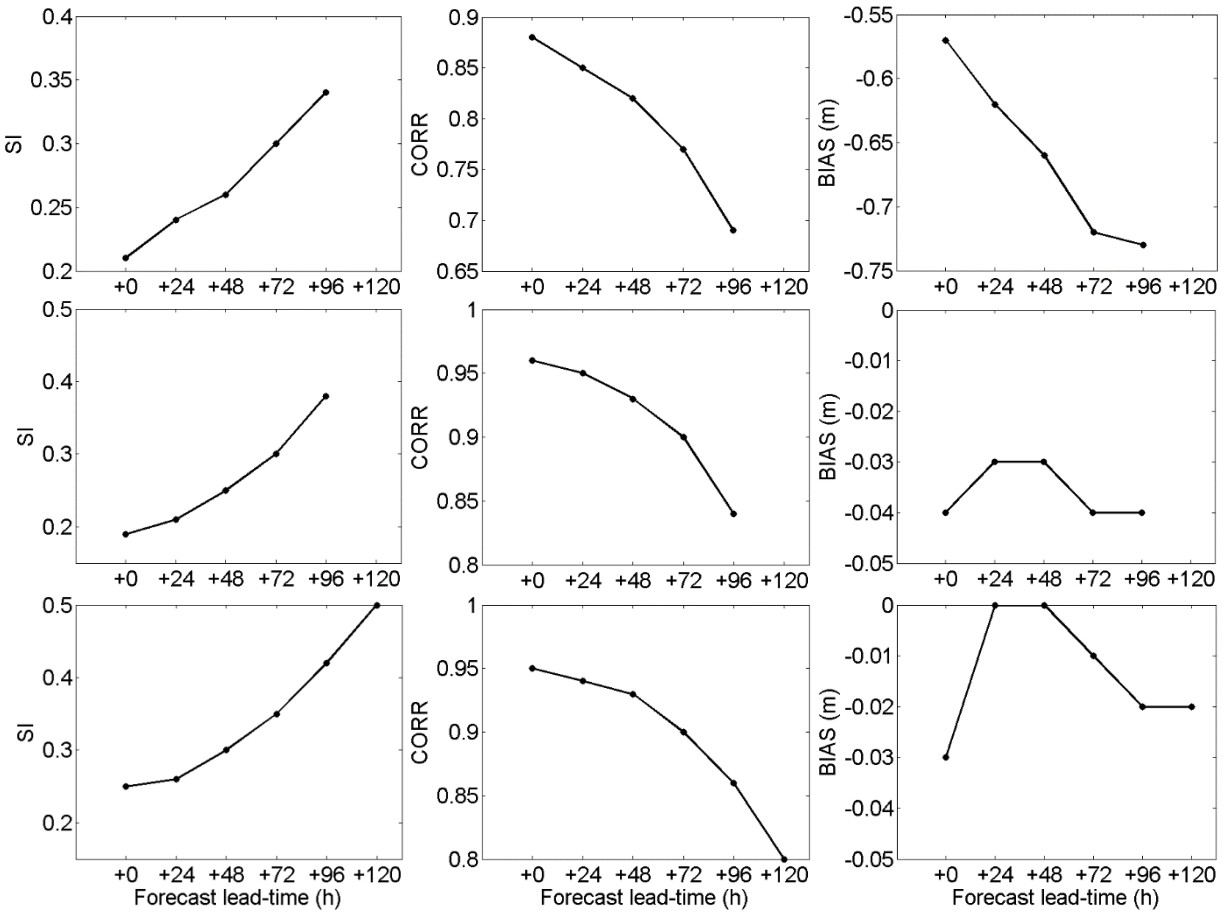

**Figure 10: ECMWF U10 forecast skill evaluated against satellite observations (top row) and Med-waves Hs forecast skill evaluated against satellite (middle row) and buoy (bottom row) observations, for the full Mediterranean Sea, for 1 year period (2014).**

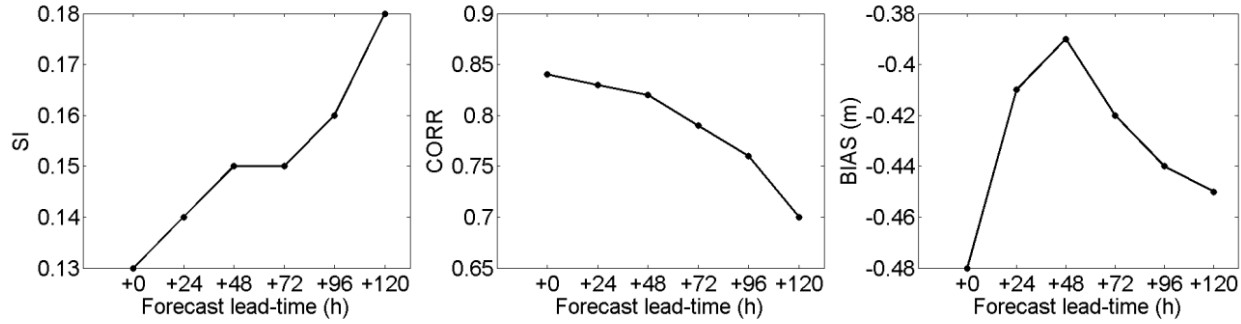

**Figure 11: Med-waves Tm forecast skill evaluated against buoy observations, for the full Mediterranean Sea, for 1 year period (2014).**