# Peer review of "Implementation and validation of a new operational wave forecasting system of the Mediterranean Monitoring and Forecasting Centre in the framework of the Copernicus Marine Environment Monitoring Service"

_Natural Hazards and Earth System Sciences, 2018_

## Referee Comment (RC1) · Dr Bidlot (Referee) · 22 Feb 2018

Not much to say, the paper explains well what was done to validate this new operational wave forecasting system, now part of CMEMS. So it will be a good reference. Bur there is not much to get excited about as the paper confirms what is already known about the different components of the system.

Specific comments: p5, line 14: 24 directions is a bit low for 1/24 degree

p5, line 18: as mentioned in the ECMWF documentation: https://www.ecmwf.int/en/elibrary/17739-part-vii-ecmwf-wave-model The value now used by ECWAM are Cdis=1.33 and delta=0.5, changes that were made as part of CY38R1, but it is also combined with a change in wind input source term, where ZALP was reduced from 0.011 to 0.008 (see (3.5 and the following discussion). This change had the desire impact of reducing low frequency energy. In the paper, it is mentioned that the North Atlantic model has too much swell. So I wonder if the change to Cdis and delta was or not accompanied with the related change to ZALP. Incidentally, as far as I can see, these CY38R1 changes were also made in WAM 4.6.2

p6, line 3: is 6-hourly forecast still the case after day 3. ECMWF outputs forecast every 3 hours up to day 6 (Actually hourly up to t+90 hours, but the data are not available to CMEMS.ECMWF should be convinced hourly forcing data would be extremely beneficial, in particular for areas such as the Mediterranean Sea. The present study highlights the difficulty of getting good wind fields for the area, and mentions a need for higher spatial resolution but it should be mentioned that temporal resolution is also essential. Note that as far as the spatial resolution, ECMWF high resolution forecasts have now since spring 2016, a ~9 km resolution (from ~ 16km used in this study

p6, line 33 ans section 4.2: Tm: are you sure you use the same spectral definition for both the model and the buoy mean wave period. WAM usually add a high frequency tail for the calculation of the all integrated parameters and hence extend all calculation of infinity, whereas buoy parameters are usually only computed to the cut-off frequency (~ 0.5 Hz). For mean wave period, such as the T02, it can results in bias (model-buoy) ~ -0.5 s So what as done?

p9, line 1: I notice that the positions for buoy 61218 and 61220 in Figure 1, do not seems to be what I have from the GTS data we have received: I have the following 61218: 43.83N, 13.72E 61220 45.33N, 12.52E

Noting the lack of agreement for 61218, even though it appears to be one of the most

exposed buoys is still surprising, so I just wonder if there is simply and position error

p11, line 23: why are there not altimeter below ∼ 2m/s ?

Minor corrections: p2, line 18: Forecast -> Forecasts mid-range -> medium-range p3, line 29: Holthuijsen p6, line 21: Stokes p8, line 21, Figure 7 -> 8 (?) p8, line 23: what is said in the text about Figure 4 is not very visible. Could you add a plot of just the QQ plot p8, line 26: there are only a few outliers, out of ∼67000 collocations, it will only have a very minor impact. p9, line 9: well reproducing -> reproducing well p11, line 17: well approximate -> approximate well

---

## Referee Comment (RC2) · Anonymous Referee #2 · 3 Apr 2018

The paper is rather well organized and well-written, supported by interesting data. It could deserve publication on NHESS provided some further aspects are discussed and cleared. Most of all, I think that better acknowledgements or pin-pint to existing publications referring to the atmosphere-ocean-wave interactions should be given. At the moment, this is just very briefly mentioned in the Introduction (line 9). This part remains indeed in the shadow; however, although this is surely not the core of the MS discussion, it should be clearly stated that fundamental steps have been taken into the

direction of "coupled" modeling. I will then address to some possible references concerning the Adriatic sea (where the model performs at its minimum) and other regions, since course similar efforts exist also in other regions of the Mediterranean. The fact that this MS is using off-line currents is a step into the right direction, but it does not exempt the authors to mention that this should follow a real coupled approach. There are, therefore, two main aspects that need to be well recalled in this work. a) The need of a real, full two-way coupling has to be recalled and well stated b) $1/16°$ is probably a too coarse resolution to expect improvements from the wave-currents feedbacks!

The authors justify the less-good performances of the model in enclosed basins and near the coast, calling for unresolved topography and fetch limitations. I would recommend some more details on the bathymetry chosen by the modle, since there exist several efforts to provide a higher-resolution bathymetry of this region (see for ap ossible check the EMODnet portal)

In general, the new forecast system provides reliable forecasts. However, model performances appear to be better in winter rather than summer, since in winter "the wave conditions are well-defined". What do the authors mean exaclty by this? Could this again be linked to the specific metocean conditions? Is this valid throughtout all the regions explored? Again, I wonder if this could be explained by a lacking consideration of the oceanic and mixed layer depth area dynamics, that could be introduced by a coupled model approach.

Future improvements: authors mention data assimilation and higher resolution wind forcing. Again, no mention is done to the coupled atmosphere-ocean-wave models, although this has proven to be a not-negligible source of increased performances exaclty in semi-enclosed seas (see references at the bottom).

Moreover, I would welcome few strategic lines discussing the scenarios under plausible climate change in the enxt decades (also with this respect I have suggested some readings to the authors)

As I stated above, the MS is too much focused on the "pure wave forecast". I think the MS would benefit a lot from an approach showing that ocean-atmosphere and waves are actually connected in a delicate interplay of energetic exchange and feedbacks. I therefore recommend to modify the Introduction and Discussion with the aim of mentioning already existing *coupled* (not just off-line as used) numerical model approaches, where the global physics of A-O-W is actually taken into account. Below here I suggest some reading/references that may be mentioned in the MS.

For improving the introduction, where possible effects of appropriate or un-accurate wave modeling efforts mya have on the overall Adriatic basin dynamics: Boldrin A. et al., 2009. The effect of Bora wind on physical and bio-chemical properties of stratified waters in the Northern Adriatic. Journal of Geophysical Research – Ocean, 114, C08S92. DOI:10.1029/2008JC004837 Bonaldo D. et al., 2015. Modelling wave-driven sediment transport in a changing climate: a case study for Northern Adriatic sea (Italy). Regional Environmental Change, 15(1), 45-55, DOI: 10.1007/s10113-014-0619-7

For mentioning the relevance that coupled approaches may have in forecasting waves in the Adriatic sea, going therefore beyond the pure wind-wave relation: Carniel S. et al., 2016. Scratching beneath the surface when coupling atmosphere, ocean and waves: analysis of a dense-water formation event. Ocean Modelling, 101, 101-112. DOI: 10.1016/j.ocemod.2016.03.007 and references therein incluided Ricchi A. et al., 2016. On the use of a coupled ocean-atmosphere-wave model during an extreme Cold Air Outbreak over the Adriatic Sea. Atmospheric Research, 172-173, 48-65. DOI: 10.1016/j.atmosres.2015.12.023

For a recent assessment of wind storminess under climate change scenarios: Bonaldo D. et al., 2018. Wind storminess in the Adriatic Sea in a climate change scenario. Acta Adriatica 58(2), 195-208
* * *

---

## Author Response (AR1)

Not much to say, the paper explains well what was done to validate this new operational wave forecasting system, now part of CMEMS. So it will be a good reference. Bur there is not much to get excited about as the paper confirms what is already known about the different components of the system.

The authors would like to thank Dr. Bidlot for the constructive review that has helped improve the quality of the manuscript. We do agree that the content of this paper is not very innovative in a research perspective. However, the description of the model and its validation is comprehensive including interesting aspects of regional validation against satellite data that have not been as thoroughly presented before. As a result, we do believe that this paper is giving a further insight regarding model performance in the Mediterranean Sea and its sub-regions and can be indeed a good reference of a CMEMS product that is totally open, free and widely used by various users ranging from research groups and port authorities to the general public. In addition, it constitutes a base reference for future upgrades of the system.

Specific comments: p5, line 14: 24 directions is a bit low for 1/24 degree

The authors take into consideration the suggestion of the reviewer that the choice of 24 directions to discretize the wave spectrum is a bit low for a wave model with a 1/24 degree spatial resolution. However the present selection of the number of frequency and directional bins was dictated by the need of setting up an operational system (and its backup) able to produce wave simulations, 5-10 days forecasts and multiyear re-analysis at 1/24 horizontal resolution given the computational resources available. In any case, in many other recent applications of spectral wave models in areas with complex wave conditions, even in higher spatial resolution, (Emmanouil G. et al. 2016, Inghilesi R. et al. 2016, Galanis G. et al. 2017, Staneva J. et al. 2017) the wave spectrum was also discretized using 24 directional bins providing sufficient reliability. However, despite the results of Piche S. et al. (2015) that show that the increase of directional bins have significant effect on computational resources for only moderate improvement in model accuracy, we plan in the future to investigate the impact of increasing the number of directional bins on the forecasting skill and if notable to upgrade our system accordingly.

p5, line 18: as mentioned in the ECMWF documentation: https://www.ecmwf.int/en/elibrary/17739-part-vii-ecmwf-wave-model The value now used by ECWAM are Cdis=1.33 and delta=0.5, changes that were made as part of CY38R1, but it is also combined with a change in wind input source term, where ZALP was reduced from 0.011 to 0.008 (see (3.5 and the following discussion). This change had the desire impact of reducing low frequency energy. In the paper, it is mentioned that the North Atlantic model has too much swell. So I wonder if the change to Cdis and delta was or not accompanied with the related change to ZALP. Incidentally, as far as I can see, these CY38R1 changes were also made in WAM 4.6.2

We have performed a comprehensive tuning of the free parameters of the dissipation function of WAM model (Cdis and delta) in order to alleviate the problem of underestimation of wave heights originating to a great extent by the underestimated surface winds, especially in enclosed areas of the Mediterranean basin. Hence, for the Mediterranean wave model the modification of these parameters (for Cdis from 2.1 to 1.33 and for delta from δ=0.6 то 0.5) seems to improve the underestimation of wave height and yields to a

better agreement with the observations (fig 1). For the coarse model, we agree with the reviewer that this combination of values for the dissipation parameters led to a slight overestimation of wave heights but we think that our system validation results are in agreement with similar studies carried out in the North-West Atlantic (Aouf et. al . 2017, Lorente et. al. 2018). Within the framework of planned forecasting system upgrades we currently test the latest available version of WAM (WAM 4.6.2) which incorporates, as the reviewer pointed out, besides these adjustments of the tuning parameters of the dissipation source term, the change of wave age tuning parameter ZALP from 0.011 to 0.008. The relevant sensitivity results will be presented in future work.

[Figure]

Fig1.

p6, line 3: is 6-hourly forecast still the case after day 3. ECMWF outputs forecast every 3 hours up to day 6 (Actually hourly up to t+90 hours, but the data are not avail-able to CMEMS.ECMWF should be convinced hourly forcing data would be extremely beneficial, in particular for areas such as the Mediterranean Sea. The present study highlights the difficulty of getting good wind fields for the area, and mentions a need for higher spatial resolution but it should be mentioned that temporal resolution is also essential. Note that as far as the spatial resolution, ECMWF high resolution forecasts have now since spring 2016, a ~9 km resolution (from ~16km used in this study

We totally agree with the reviewer that in wind-wave dominated areas with complex orography such as the Mediterranean Sea, the accuracy of the wave model is strongly dependent on the quality and resolution (spatial and temporal) of the wind forcing used to drive model analyses and forecasts. Considering that our system is fully aligned to all the latest ECMWF outputs operationally available, higher temporal and spatial resolution wind fields offered by ECMWF would be a significant add on allowing the wave model to better capture the wave variability in the Mediterranean Sea.

As the reviewer suggested, in the revised manuscript, we have added some text highlighting the importance of the temporal resolution of the wind forcing. In particular, this is done in p6* (Section 2), lines 7-12 and in p17, lines 15-16 (Section 5).

*Note that all references to manuscript pages and numbers refer to the annotated revised manuscript included in this pdf.

p6, line 33 ans section 4.2: Tm: are you sure you use the same spectral definition for both the model and the buoy mean wave period. WAM usually add a high frequency tail for the calculation of the all integrated parameters and hence extend all calculation of infinity, whereas buoy parameters are usually only

computed to the cut-off frequency (~ 0.5 Hz). For mean wave period, such as the T02, it can results in bias (model-buoy) ~ -0.5 s So what as done?

We have used the Tm02 mean wave period for both the model results and the buoy observations. The computation employed for the model estimation of Tm02 includes the high frequency spectral tail as this is standard for the WAM model. On the other hand, for the wave buoys, the frequency band (and the frequency cut off) where the mean wave period is calculated depends on the wave sensor and the software employed by each data provider and can vary for different buoys over the Mediterranean Sea (INSITU TAC, personal communication). Thus, a non-standard model estimation of Tm02 for each wave buoy over the basin, can be quite complicated. As a result, we believe that it is acceptable to proceed with the current approach and anticipate that there will be a bias in the mean wave period estimation.

In the revised manuscript, p7 (Section 3), lines 25-27 have been added in order to explain the difference in the computation of model and buoy Tm and the bias this difference is anticipated to introduce in the model-buoy comparisons. In line with the above, in p14 (Section 4.2), lines 29-31 have been added.

p9, line 1: I notice that the positions for buoy 61218 and 61220 in Figure 1, do not seems to be what I have from the GTS data we have received: I have the following 61218: 43.83N, 13.72E 61220 45.33N, 12.52E

The GTS position for location 61218 seems to coincide with the location of the buoy in Figure 2 (GTS and CMEMS IN-SITU TAC locations coincide in Google Earth). Regarding buoy 61220, the GTS coordinates place it more to the north than the CMEMS IN-SITU TAC coordinates which are: 44.97N, 12.66E. The latter have been used as provided by the CMEMS IN-SITU TAC; however, we have sent a request to the TAC for checking this out.

Noting the lack of agreement for 61218, even though it appears to be one of the most exposed buoys is still surprising, so I just wonder if there is simply a position error

From the previous, at least with respect to buoy 61218, we believe that this is not a position error. Firstly, the buoy is located within the North Adriatic Sea, which as shown in Figure 8, p33 (revised manuscript) is a region with poor overall statistics compared to the rest of the Mediterranean Sea. In addition, the QQ-Scatter and time-series plots below (fig. 2 & 3), which correspond to location 61218 for year 2014, show that the model has failed to reproduce a number of stormy wave events at the location. These 'outliers' (also in Figure 4, p30, revised manuscript) could have introduced some extra bias to the computed statistics. These events happened between the 10[th] and 15[th] of January 2014 and mostly belong to a single storm.

[Figure]

Fig2

[Figure]

Fig. 3

p11, line 23: why are there not altimeter below ~ 2m/s?

This came from an older analysis where altimeter U10 obs < 2 m/s have been removed considering that altimeter measurements below 2 m/s are not very reliable (e.g. Cavaleri and Sclavo, 2006). However, in later analysis and now in the paper (revised version) we have included all observations obtained from CERCAT-IFREMER considering that these are filtered and corrected and according to the provider they can all be considered as valid. The revised analysis has lead to marginal improvement in BIAS and CORR and a marginal deterioration in SI. In general, full MED and regional statistics have been so slightly affected by this revision.

The revision includes:
- p32, Fig. 7 (left)
- p12, lines 6-7
- p33, Fig. 8 (left column)
- p35, Fig. 10 (top row)
- p16, lines 9, 15 (change of percentage numbers)

Minor corrections:
p2, line 18: Forecast -> Forecasts mid-range -> medium-range
Corrected
p3, line 29: Holthuijsen
Corrected (p3, line 30)
p6, line 21: Stokes
Corrected (p7, line 6)
p8, line 21, Figure 7 -> 8 (?)
It was initially Fig. 8, corrected to Fig. 7 (p9, line 11)
p8, line 23: what is said in the text about Figure 4 is not very visible. Could you add a plot of just the QQ plot
p30, Figure 4: the QQ plot alone has been added.
p8, Section 4.1.1, first paragraph: the figure description has been modified in agrrement.
p8, line 26: there are only a few outliers, out of ~67000 collocations, it will only have a very minor impact.
These outliers all correspond to location 61218 in the Adriatic Sea as seen in the Scatter plot inserted above.
In the revised manuscript, p9, lines 6-20, this fact is mentioned while the comment on the impact of the outliers on the overall statistics is removed.
p9, line 9: well reproducing -> reproducing well
Corrected (p9, line 32)
p11, line 17: well approximate -> approximate well
Corrected (p11, line30)

Author(s) 2018. This work is distributed under the
Creative Commons Attribution 4.0 License.

[Figure]

The paper is rather well organized and well-written, supported by interesting data. It could deserve publication on NHESS provided some further aspects are discussed and cleared. Most of all, I think that better acknowledgements or pin-pint to existing publications referring to the atmosphere-ocean-wave interactions should be given. At the moment, this is just very briefly mentioned in the Introduction (line 9). This part remains indeed in the shadow; however, although this is surely not the core of the MS discussion, it should be clearly stated that fundamental steps have been taken into the direction of "coupled" modeling.

We thank the reviewer for his/her interesting and helpful comments which have all been taken into account and answered. In the revised version of the manuscript, we have enriched the introduction and the conclusions with text related to atmosphere – ocean – wave interactions through a full coupling indicating possible impacts on wave model performance.

More specifically, related changes* can be found in:
- p3, line 20: addition of a reference
- p4, lines 5-18: discussion, including references, on the importance of full-coupling and of the associated scientific and operational challenges
- p17, lines 13-23: additional conclusions considering full coupling potential

*Note that all references to manuscript pages and numbers refer to the annotated revised manuscript included in this pdf.

I will then address to some possible references concerning the Adriatic sea (where the model performs at its minimum) and other regions, since course similar efforts exist also in other regions of the Mediterranean.

As mentioned above, in the introduction and in the conclusions of the manuscript we have taken into account suggested work adding appropriate references, including some of the references the reviewer suggested.

See above for pages and lines where changes have been introduced.

The fact that this MS is using off-line currents is a step into the right direction,

but it does not exempt the authors to mention that this should follow a real coupled approach. There are, therefore, two main aspects that need to be well recalled in this work. a) The need of a real, full two-way coupling has to be recalled and well stated b) 1/16° is probably a too coarse resolution to expect improvements from the wave-currents feedbacks!

a) In the conclusions of the manuscript we mention that one of our near future goals in the framework of CMEMS is the two-way coupling between the oceanic and wave component of the Mediterranean MFC. This is a collaborative approach between the CMCC scientific team who leads the modelling of the physical part and the HCMR team who is responsible for wave forecasting within MED MFC. Preliminary results of this work are shown in Clementi et. al. (2017).

In the revised manuscript relevant additions and modifications can be found in p18, lines 18-23.

b) The following figure (fig1) shows the Symmetrically Normalized RMSE (SNRMSE), which has been found to be more robust than RMSE when comparing data with similar level of performance (Hanna and Heinold, 1985), between wave buoy observations at individual wave buoy locations (red dots) and 1/24° (top numbers) and 1/16° (bottom numbers) surface currents forcing respectively, for 1 year period (2014). Top numbers highlighted in green indicate a better metric value compared to bottom numbers (i.e. 1/24 better than 1/16). Otherwise, bottom numbers correspond to a better or equal metric value. The figure shows that a small improvement (< 2%) occurred with the increased resolution surface currents forcing at 20 out of the 31 buoys examined. The increase of surface currents resolution from 1/16° to 1/24° pertains to the following version of the CMEMS MED MFC Waves System.

[Figure]

*Fig.1 Off-line coupling with 1/24 surface currents does not improve the scores in the Adriatic wave buoys examined*

The authors justify the less-good performances of the model in enclosed basins and near the coast, calling for unresolved topography and fetch limitations. I would recommend some more details on the bathymetry chosen by the model, since there exist several efforts to provide a higher-resolution bathymetry of this region (see for a possible check the EMODnet portal)

The problem of "unresolved topography" refers to the inability of the relative low resolution (1/24°) wave forecasting system to resolve fine bathymetric features,

encountered, for example, in regions such as the Adriatic and Aegean Seas. In addition, it refers to the inability of the relative low resolution (1/8º) ECMWF wind forcing to properly resolve fine orography. Consequently, the problem of unresolved bathymetry cannot merely be tackled by a higher resolution bathymetry (e.g. EMODNET) without the synchronous increase of wave model resolution. This would only lead to insignificant change in the performance of the system in marine areas of complex topography. Moreover, for a major improvement of the wave model performance an increase of wind forcing resolution would be required.

In general, the new forecast system provides reliable forecasts. However, model performances appear to be better in winter rather than summer, since in winter "the wave conditions are well-defined". What do the authors mean exactly by this? Could this again be linked to the specific metocean conditions? Is this valid throughout all the regions explored? Again, I wonder if this could be explained by a lacking consideration of the oceanic and mixed layer depth area dynamics, that could be introduced by a coupled model approach.

As explained in the manuscript (p9, lines 3-4, revised manuscript), well-defined wave conditions refer to stormy conditions with well-defined patterns and higher waves. In other words, well-defined wave conditions are linked to strong winds which lead to a more organized sea with better-formed waves. This is a common result in the literature (references are provided in p9, lines 4-5) and is mostly associated to a better quality wind forcing in 'stormy' conditions. For example, Ardhuin et al (2007), which examined wind and wave model performance in the western Mediterranean Sea using four difference sources of wind models (including ECMWF) to force three different wave models, concluded:"The accuracy of the model wind fields depends on how well the meteorological situation is defined. In stormy, well-extended areas the models are more consistent to each other. For more uncertain situations the percent errors tend to be larger". Also, the same study found that wave model forecast skill deteriorates for low waves (Hs < 1.5 m), particularly close to the coasts. In our study, enhanced winter wave model performance is valid for all the regions explored except for the central and east Levantine Basin (lev2, lev3, lev4). Despite the difficulties in the direct comparison of wind and wave model performance due to the change of the whitecapping dissipation coefficients in WAM (see p12, second paragraph) which strongly offsets bias, we found that a considerably reduced overall wind performance is observed in the Levantine Basin in winter in relation to summer with the SNRMSE being higher by 5% - 13% in the former season. This could explain why the wave model performance in the Levantine Basin in worst in winter than in summer whilst the contrary is true in all other regions.

We believe that the current version of our wave system includes all the mechanisms with an important role in modifying the wave field at the current horizontal resolution. Sea-level variations during extreme weather conditions could play a significant role in modifying the wave field, however, this is true very near the shore where a coastal wave model application would be appropriate. Wave-current full 2-way coupling could be a possible mechanism for improving wave model results although we believe that the 2-way coupling approach is more beneficial for the circulation (e.g. wave modified surface stress) and hydrology (e.g. wave induced vertical mixing, additional advection through the stokes drift velocity) than for the waves.

Future improvements: authors mention data assimilation and higher resolution wind forcing. Again, no mention is done to the coupled atmosphere-oceanwave models, although this has proven to be a not-negligible source of increased performances exactly in semi-enclosed seas (see references at the bottom).

Moreover, I would welcome few strategic lines discussing the scenarios under plausible climate change in the next decades (also with this respect I have suggested some readings to the authors)

The authors find it hard to see the relationship between the performance of an operational analysis and forecast wave system and wave climate scenarios or wave climate statistics of any nature. Hindcast or reanalysis wave products, also included within the CMEMS framework, would be suited for such a correlation.

As I stated above, the MS is too much focused on the "pure wave forecast". I think the MS would benefit a lot from an approach showing that ocean-atmosphere and waves are actually connected in a delicate interplay of energetic exchange and feed-backs. I therefore recommend to modify the Introduction and Discussion with the aim of mentioning already existing *coupled* (not just off-line as used) numerical model approaches, where the global physics of A-O-W is actually taken into account. Below here I suggest some reading/references that may be mentioned in the MS.

As it is clear by now from our replies to the previous comments of the reviewer, the introduction and the conclusions of the manuscript have been considerably modified with the aim of mentioning already existing literature on A-O-W coupling, including some of the references suggested by the reviewer. In addition, the benefits and the difficulties of a full coupling system for an operational wave forecast system in the Mediterranean Sea have been outlined.

For improving the introduction, where possible effects of appropriate or un-accurate wave modeling efforts mya have on the overall Adriatic basin dynamics: Boldrin A. et al., 2009. The effect of Bora wind on physical and bio-chemical properties of strati-fied waters in the Northern Adriatic. Journal of Geophysical Research – Ocean, 114, C08S92. DOI:10.1029/2008JC004837 Bonaldo D. et al., 2015. Modelling wave-driven sediment transport in a changing climate: a case study for Northern Adriatic sea (Italy). Regional Environmental Change, 15(1), 45-55, DOI: 10.1007/s10113-014-0619-7

For mentioning the relevance that coupled approaches may have in forecasting waves in the Adriatic sea, going therefore beyond the pure wind-wave relation: Carniel S. et al., 2016. Scratching beneath the surface when coupling atmosphere, ocean and waves: analysis of a dense-water formation event. Ocean Modelling, 101, 101-112. DOI: 10.1016/j.ocemod.2016.03.007 and references therein inciuded Ricchi A. et al., 2016. On the use of a coupled ocean-atmosphere-wave model during an extreme Cold Air Outbreak over the Adriatic Sea. Atmospheric Research, 172-173, 48-65. DOI: 10.1016/j.atmosres.2015.12.023

For a recent assessment of wind storminess under climate change scenarios: Bonaldo D. et al., 2018. Wind storminess in the Adriatic Sea in a climate change scenario. Acta Adriatica 58(2), 195-208

[revised manuscript text omitted]
 is not presently coupled with an ocean model (coupling is done only with the wave component), does not consider some vigorous air-sea interactions processes (large heat fluxes or strong ocean mixing processes) that occur in regions which are usually affected by extreme weather events such as the northern part of the Adriatic during strong wind

20  events (Bora, Sirocco) and, as a result, it fails to properly reproduce the spatial structure of the wind fields. To overcome this limitation, many studies (Carniel et al., 2016; Ricchi et al. 2016, 2017) show that the use of fully coupled atmosphere – ocean – wave model can be considered appropriate for these regions for properly representing the air–sea interactions and for providing a more realistic and consistent evolution of the atmospheric and oceanic fields.

The mean wave period is reasonably well simulated by the model. The RMSE is 0.7 s and is mainly caused by

25  model bias which has a value of -0.48 s (12%). In general, the model underestimates the observed mean wave period and exhibits greater variability than the observations. A relatively larger model underestimate is found for mean wave periods below 4.5 s. The scatter index is 13%, the correlation coefficient is 0.85 and the best-fit slope is 0.88. Model performance is a little better in winter when wave conditions are well-defined. Spatially, the model somewhat overestimates the highest mean wave period values in the western Mediterranean Sea, west and south of France. Otherwise, model underestimate is

30  widespread. Similarly to the wave height, the model performance is best at well-exposed offshore locations and deteriorates near the shore mainly due to fetch limitations.

The forecast skill of the model over the Mediterranean Sea deteriorates with forecast range. The growth of error in the wave forecast is mainly due to the growth of error in the forcing wind fields. The scatter index of the significant wave height deteriorates by 19% and 25% over the 5-day forecast for model-satellite and model-buoy comparisons respectively.

The equivalent deterioration for mean wave period is only 5% (model-buoy comparison). A monotonic decrease in correlation is also observed. On the contrary, the evolution of bias with forecast range shows some variability with no clear trend. Nevertheless, this variability does not exceed 3% over the forecast period.

In the near future an Optimal Interpolation type data assimilation scheme will be added to the Med-waves system in order to blend satellite along-track significant wave height measurements with model background forecasts. Although wave data assimilation is known not to be particularly beneficial in areas where wind sea conditions are dominant we expect that wave forecasts in certain sub-areas of the Mediterranean Sea where swell propagation is quite frequent, will be improved at +24h and perhaps +48h lead time. The enhanced Med-waves system with the data assimilation system module is going to produce 3-hourly wave analyses on a daily basis for the Mediterranean Sea by assimilating Sentinel-3 and Jason-3 altimeter measured significant wave heights and surface winds. The assimilation will be based on the inherent data assimilation scheme of WAM Cycle 4.5.4 model which generates an updated wave field by distributing the information from the observed significant wave height and surface wind data within a given time window over the entire model grid. The Med-waves Data Assimilation component is already integrated into the Med-waves system  (April 2018).

Another planned  upgrade to the forecasting system is the future downscaling of the ECMWF atmospheric model solution so that higher  resolution wind analyses and forecasts will be available to force the  wind waves dynamics with expected  
[revised manuscript text omitted]

---

## Author Response (AR2)

**Authors's response**

Contents:

**Response to Reviewer #1**

The paper demonstrate the quality of the wave modelling system and its weakness.
I have the following corrections/update

page 3, line 23: note that the operational ECMWF 10 km uncoupled wave model has been using surface currents for years. These currents were originally provided by the Norwegian TOPAZ system, which is now part of CMEMS (Bidlot, private communication)

The authors are aware of the fact that the ECMWF wave model is using surface currents. In the relevant paragraph, we focus on Mediterranean wave model implementations in specific. Moreover, we aim to highlight that no regional Mediterranean models consider both the Atlantic influence and the surface currents in their model suit.

page 6: it is worth noting that following the decision at its June 2018 council, ECMWF can make its high resolution (HRES) forecasts available every hourly (from 3-hourly) up to forecast step 90 hour.

The recent decision of ECMWF council is a step forward and will definitely impact the skill of the Med-waves forecasting system. We have revised the conclusions to include this information using Bidlot, private communication, as a reference.

Also note and comment in the conclusion, that the resolution of ECMWF HRES in 2004 was 16km, but since spring 2016, it is now 9km

The conclusions were revised to include this comment.

page 10, line 3,
long term validation of operational wave forecasts from global operational centres, has also nicely demonstrated, the difficulty of representing waves in enclosed areas, See
https://www.ecmwf.int/en/newsletter/150/meteorology/twenty-one-years-wave-forecast-verification

Thank you for the reference. Figure 3 in this article is indeed a very nice demonstration of the difficulty of representing waves in the Mediterranean Sea. We have included the reference in the introduction.

refer also, to the newly published white paper "wave modelling coastal inner seas
https://owncloud.ve.ismar.cnr.it/owncloud/index.php/s/0nIEW6wJXvkgAar
Cavaleri et al, 2018. Wave modelling in coastal and inner seas, PIO,
pp.1-70.

The reference of Cavaleri et al., 2018 has been added here and in the introduction.

page 17:
it is now official, since June 5 2018, ECMWF High Resolution system (the one used by CMEMS) is a fully coupled atmosphere-waves-ocean-sea ice system

see for instance
https://www.ecmwf.int/en/elibrary/18491-newsletter-no-156-summer-2018

The conclusions were revised to include this update.

Please revise your conclusions to indicate that major progress has been made since 2004 by ECMWF, and other major NWP centres, insofar as resolutions, both spatial and temporal, and in modelling relevant exchange progresses.

The conclusions have been revised to include updates / improvements related to the ECMWF winds used to force Med-waves.

Minor corrections:
p2, line 18: Forecast (ECMWF) -> Forecasts (ECMWF)
provides since 1992 -> has been providing since 1992

Corrected

p7, line 7: Stokes drift -> surface Stokes drift (?)

Modified (appearing as Stokes drift in the CMEMS catalogue)

p10, line 1: more close -> closer

Corrected

Table 7 : bias(m) -> bias (s)

All units in this table have been corrected

Figure 11: BIAS(m) -> BIAS(m/s) for wind and BIAS(s) for period

Corrected

**Response to Reviewer #2**

The authors have amended the manuscript, harmonizing it in the direction that I proposed in the first-round review. When they disagreed with some comments, they provided pertinent replies.

At this point, I think the MS is indeed scientifically solid and can be published. However, I feel the need to give more credit to previosuly existing seminal papers dealing with wave forecast in the Med region. I am providing some titles here below, from where the authors may extract a few lines (to be used in the Introduction?):

- Cavaleri et al (1991), Wind wave cast in the Mediterranean Sea, Journal of Geophysical Research 96:10:739-10,764
- Bolaños, R et al (2005), Limits of operational wave prediction in the North-Western mediterranean,Proceedings of the Coastal Engineering
- Bertotti and Cavaleri (2009),Large and small scale wave forecast in the Mediterranean, nhess, Volume 9 Pages 779-788

All the suggested references have been added in the introduction.

Also, a minor point, note that the correct citation to the Ricchi et al. 2017 paper is:

[revised manuscript text omitted]
 | $\bar{R}$ (ms) | $\bar{M}$ (ms) | STD R (ms) | STD M (ms) | RMSE (ms) | SI | BIAS (ms) | CORR | SLOPE |
|---|---|---|---|---|---|---|---|---|---|---|
| 61198 | 2447 | 3.76 | 3.56 | 0.76 | 0.89 | 0.47 | 0.11 | -0.20 | 0.88 | 0.95 |
| 61417 | 2740 | 3.97 | 3.72 | 0.75 | 0.86 | 0.47 | 0.10 | -0.25 | 0.89 | 0.94 |
| 61281 | 2125 | 3.59 | 3.28 | 0.65 | 0.78 | 0.53 | 0.12 | -0.31 | 0.84 | 0.92 |
| 61280 | 2306 | 3.71 | 3.30 | 0.66 | 0.74 | 0.58 | 0.11 | -0.41 | 0.84 | 0.89 |
| 61430 | 2500 | 4.20 | 3.71 | 0.87 | 1.06 | 0.68 | 0.11 | -0.50 | 0.90 | 0.89 |
| 61197 | 2748 | 4.28 | 4.19 | 1.19 | 1.21 | 0.63 | 0.15 | -0.09 | 0.86 | 0.97 |
| 61196 | 2890 | 4.41 | 3.79 | 0.79 | 0.92 | 0.75 | 0.10 | -0.62 | 0.88 | 0.86 |
| 61188 | 2028 | 3.53 | 3.03 | 0.75 | 0.77 | 0.69 | 0.13 | -0.50 | 0.81 | 0.86 |
| 61191 | 1800 | 3.45 | 2.92 | 0.80 | 0.85 | 0.67 | 0.12 | -0.53 | 0.88 | 0.85 |
| 61190 | 1773 | 3.42 | 2.92 | 0.87 | 0.87 | 0.67 | 0.13 | -0.50 | 0.88 | 0.85 |
| 61431 | 621 | 3.82 | 3.10 | 0.78 | 0.84 | 0.88 | 0.13 | -0.72 | 0.81 | 0.81 |
| 61289 | 2157 | 3.82 | 3.30 | 0.70 | 0.78 | 0.66 | 0.11 | -0.52 | 0.85 | 0.87 |
| 61021 | 1531 | 4.32 | 3.77 | 0.90 | 0.98 | 0.76 | 0.12 | -0.55 | 0.85 | 0.88 |
| 61187 | 1410 | 4.33 | 3.33 | 1.00 | 0.80 | 1.26 | 0.18 | -1.00 | 0.65 | 0.76 |
| 61295 | 802 | 3.66 | 2.97 | 0.75 | 0.79 | 0.82 | 0.12 | -0.69 | 0.83 | 0.81 |
| 68422 | 2008 | 4.26 | 3.55 | 0.87 | 1.07 | 0.85 | 0.11 | -0.71 | 0.90 | 0.84 |
| 61277 | 2112 | 4.02 | 3.45 | 0.68 | 0.81 | 0.70 | 0.10 | -0.56 | 0.85 | 0.86 |
| SARON | 1809 | 3.18 | 2.75 | 0.45 | 0.60 | 0.60 | 0.13 | -0.43 | 0.72 | 0.87 |
| ATHOS | 1440 | 3.86 | 3.10 | 0.70 | 0.77 | 0.83 | 0.09 | -0.76 | 0.89 | 0.81 |

[Figure]

$$\left\{\frac{\partial}{\partial t} + \overrightarrow{c_g}\ \frac{\partial}{\partial \overrightarrow{x}}\right\} N(\vec{x}, t, f, \theta) = S_{in} + S_{dis} + S_{NL} + S_{bot}$$

Figure 1: Schematic of the Med-waves system.

[Figure]

Figure 2: Wave buoys' location and unique ID code.

[Figure]

**Figure 3: Mediterranean Sea sub-regions for qualification metrics.**

[Figure]

**Figure 4: QQ-Scatter plots (left) and QQ plot alone (right) of Med-waves output Hs versus wave buoys' observations, for the full**
5 **Mediterranean Sea, for 1 year period (2014): QQ-plot (black crosses), 45° reference line (dashed red line), least-squares best fit line**
**(red line, left plot).**

[Figure]

**Figure 5: QQ-Scatter plots of Med-waves output Hs versus wave buoy observations at specific wave buoy locations, for 1 year period (2014): QQ-plot (black crosses), 45° reference line (dashed red line), least-squares best fit line (red line).**

[Figure]

**Figure 6: Med-waves Hs evaluation against satellite Hs, for each Mediterranean Sea sub-region shown in Fig. 3, for 1 year period (2014).**

[Figure]

**Figure 7: QQ-Scatter plots of: (left) ECMWF forcing wind speed U10 versus satellite U10 (Jason-2); (right) Med-waves Hs versus satellite Hs (Jason-2 and Saral), for the full Mediterranean Sea, for 1 year period (2014).**

[Figure]

**Figure 8: ECMWF U10 (left column) and Med-waves Hs (right column) evaluation against satellite U10 (Jason-2) and satellite Hs (Jason-2 and Saral) respectively: maps of metric values over the Mediterranean Sea sub-regions shown in Fig. 3, for 1 year period (2014).**

[Figure]

**Figure 9: QQ-Scatter plots of Med-waves output versus wave buoys' observations, for the full Mediterranean Sea, for 1 year period (2014).**

[Figure]

[Figure]

**Figure 10: ECMWF U10 forecast skill evaluated against satellite observations (top row) and Med-waves Hs forecast skill evaluated against satellite (middle row) and buoy (bottom row) observations, for the full Mediterranean Sea, for 1 year period (2014).**

[Figure]

**Figure 11: Med-waves Tm forecast skill evaluated against buoy observations, for the full Mediterranean Sea, for 1 year period (2014).**